# Laser induced fluorescence based detection of atmospheric nitrogen dioxide and comparison of different techniques during the PARADE 2011 field campaign

Umar Javed[1,2], Dagmar Kubistin[1,3,4], Monica Martinez[1], Jan Pollmann[1], Markus Rudolf[1], Uwe Parchatka[1], Andreas Reiffs[1], Jim Thieser[1], Gerhard Schuster[1], Martin Horbanski[5], Denis Pöhler[5], John N. Crowley[1], Horst Fischer[1], Jos Lelieveld[1], and Hartwig Harder[1]

[1]Department of Atmospheric Chemistry, Max Planck Institute for Chemistry, Mainz, Germany
[2]Institute of Energy and Climate Research, IEK-8: Troposphere, Forschungszentrum Jülich GmbH, Jülich, Germany
[3]University of Wollongong, School of Chemistry, Wollongong, NSW, Australia
[4]German Meteorological Service, Meteorological Observatory Hohenpeissenberg (MOHp), Hohenpeissenberg, Germany
[5]Institute of Environmental Physics, University of Heidelberg, Heidelberg, Germany

*Correspondence to*: Hartwig Harder (hartwig.harder@mpic.de) or Umar Javed (u.javed@fz-juelich.de)

**Abstract.** GANDALF (**G**as **A**nalyzer for **N**itrogen **D**ioxide **A**pplying **L**aser-induced **F**luorescence), a new instrument for the detection of nitrogen dioxide based on the laser-induced fluorescence (LIF) technique, is presented in this paper. GANDALF is designed for ground based and air-borne deployment with a robust calibration system. In the current setup, it uses a multi-mode diode laser (447 – 450 nm) and performs in situ, continuous, and autonomous measurements with a laser pulse repetition rate of 5 MHz. The performance of GANDALF was tested during the summer of year 2011 (15 Aug.-10 Sep.) in a field experiment at Kleiner Feldberg, Germany. The location is within a forested region with urban influence where $NO_x$ levels were between 0.12 and 22 parts per billion by volume (ppb). Based on the field results, the limit of detection is estimated at 5 – 10 parts per trillion by volume (ppt) in 60 s at a signal to noise ratio (SNR) of 2. The overall accuracy and precision of the instrument are better than 5 % (1 σ) and 0.5 % + 3 ppt (1 σ min$^{-1}$), respectively. A comparison of nitrogen dioxide measurements based on several techniques during the field campaign PARADE-2011 is presented to explore methodic differences.

## 1 Introduction

Tropospheric nitric oxide (NO) and nitrogen dioxide ($NO_2$) are key species in atmospheric chemistry and are strongly coupled due to their fast photochemical interconversion generally combined as $NO_x$ (= $NO + NO_2$). Nitrogen oxides act as key catalyst in the formation of tropospheric ozone ($O_3$) (Crutzen, 1979). $NO_x$ also plays an important role in the oxidation capacity of the troposphere by affecting the abundances of $O_3$, hydroxyl radical (OH), and nitrate radical ($NO_3$).

The main sources of $NO_x$ in the troposphere are combustion processes, predominantly fossil fuel use, biomass burning, microbial production in soils, transport from the stratosphere and lightning, the latter two directly affecting the free troposphere [e.g. (Logan, 1983)] along with aircraft emissions (Strand and Hov, 1996). $NO_x$ emissions from the surface are mostly in the form of NO which is converted to $NO_2$ by the reaction of NO with $O_3$, the hydroperoxyl radical ($HO_2$), organic peroxy radicals ($RO_2$), and halogen oxides. The oxidation of nitrogen oxides in the atmosphere leads to the formation of several reactive nitrogen species, some of which act as reservoirs for $NO_x$, denoted by $NO_z$[1]. The $NO_x$ lifetime is largely determined by its oxidation into nitric acid ($HNO_3$) by OH during daytime, and in polluted air also by the heterogeneous loss of $N_2O_5$ (formed by $NO_2+NO_3$) on

---

[1] $NO_z = NO_3 + 2N_2O_5 + HNO_3 + HONO + RO_2NO_2 + RONO_2 + HNO_4 + $ Particulate Nitrate + …

wet surfaces during the night, e.g. on aerosols and cloud droplets. The tropospheric lifetime of $NO_x$ is in the range of hours to days and it is generally shorter closer to the surface of Earth compared to high altitudes [e.g. (Ehhalt et al., 1992)]. Because of its relatively short lifetime, the transport distance of $NO_x$ is limited, compared to other primary pollutants like carbon monoxide (CO) and methane ($CH_4$) that disperse on hemispheric and global scales.

The wet and dry deposition of $HNO_3$ is considered the major sink for $NO_x$. Uncertainties in the $NO_x$ budget have recently been highlighted (Stavrakou et al., 2013). These include the uncertainty in the estimate of the rate coefficient for $NO_2 + OH$ under tropospheric conditions (Mollner et al., 2010), a lack of proper representation in chemical mechanisms for the loss of $NO_x$ via organic nitrate formation (Browne and Cohen, 2012), and the formation of $HNO_3$ in a minor branch of the reaction between NO and $HO_2$ (Butkovskaya et al., 2007) which showed significant impacts on the concentration of $NO_x$, OH, $HNO_3$ and related chemistry (Cariolle et al., 2008;Gottschaldt et al., 2013). Additionally, a lack of agreement between modelled and measured OH concentrations over forests (Lelieveld et al., 2008;Kubistin et al., 2010) and urban regions (Hofzumahaus et al., 2009) contribute to uncertainty about $NO_x$ chemistry. In summary, $NO_x$ even in the low ppt range is important for understanding the tropospheric $O_3$ production (Lelieveld and Crutzen, 1990;Carpenter et al., 1997) and the cycling of radicals (Monks, 2005). Therefore, it is of great importance to have accurate $NO_x$ measurements from regional to global scales.

Tropospheric mixing ratios of $NO_x$ can vary from a few ppt to hundreds of ppb, depending on remote (Hosaynali Beygi et al., 2011) and urban conditions (Clapp and Jenkin, 2001), respectively. The high temporal and spatial variability of $NO_x$ with the wide concentration ranges challenges its measurements. Briefly, several different methods have been used to measure $NO_x$ in the atmosphere. The Photofragmentation Two-Photon Laser-Induced Fluorescence (PF-TP-LIF) (Sandholm et al., 1990;Bradshaw et al., 1999) and chemiluminescence (Fontijn et al., 1970) methods are well known for direct in situ NO detection. In the past, an indirect detection of $NO_2$ with these techniques has been performed by converting $NO_2 \rightarrow NO$ via photolytic/catalytic process followed by NO detection. However, in the case of $NO_2$ to NO conversion, a potential interference from $NO_z$ species cannot be fully excluded for the $NO_2$ measurement, e.g. (Crawford et al., 1996;Villena et al., 2012;Reed et al., 2016). Therefore, a direct detection of $NO_2$ is advantageous. Techniques like cavity ring down absorption spectroscopy (Osthoff et al., 2006), tunable diode laser absorption spectroscopy (Herndon et al., 2004), cavity enhanced absorption spectroscopy (Wojtas et al., 2007), cavity-enhanced differential optical absorption spectroscopy (Platt et al., 2009), and cavity attenuated phase shift spectroscopy (Ge et al., 2013) provide direct in situ detection of $NO_2$. Another promising method for a direct $NO_2$ detection is based on the laser-induced fluorescence technique. The LIF method for $NO_2$ provides highly selective and sensitive measurements and it has already been demonstrated successfully in the past with detection limits reaching down to about 5 ppt min$^{-1}$ (Thornton et al., 2000;Matsumoto and Kajii, 2003).

An overview of LIF $NO_2$ systems from the literature is given in Table 1. LIF systems have been used for many years but the detection limits are sometimes not suitable for detection in a remote region, especially in some of the earlier attempts (George and Obrien, 1991;Fong and Brune, 1997;Matsumoto et al., 2001;Taketani et al., 2007). In the last decade, owing to the advancements in lasers, better detection limits have been achieved. The LIF systems have shown good selectivity and sensitivity (Thornton et al., 2000;Matsumi et al., 2001;Matsumoto and Kajii, 2003;Dari-Salisburgo et al., 2009;Di Carlo et al., 2013), but most of these systems have large (typically > 50 kg) and complex laser systems. The availability of much smaller and lighter diode lasers have made it possible to build compact instruments with the caveat of lower laser power and higher detection limits. Here for GANDALF, a high power, lightweight diode laser (< 2 kg) system is used to achieve a compact design with detection limits comparable to those of the best performing larger instruments.

In the following the newly developed LIF instrument for the direct $NO_2$ detection is described. Results from the first field deployment in a semi-rural region are reported to demonstrate the performance of the instrument. Measurements of trace

gases along with meteorological parameters were carried out during the campaign, including $NO_2$ measurements based on several techniques, namely LIF, cavity ring down absorption spectroscopy, two-channel chemiluminescence detection, cavity-enhanced differential optical absorption spectroscopy, and long-path differential optical absorption spectroscopy. Being the first deployment of GANDALF, this opportunity provided the means for a detailed comparison to other methods under real atmospheric conditions.

## 2 The instrument description

### 2.1 The operational method

The measurements of GANDALF are based on laser-induced fluorescence at low pressure (<10 hPa). The $NO_2$ molecule is excited by a diode-laser with a wavelength well above the photolysis threshold ($\lambda > 420$ nm for $NO_2$) and the red-shifted fluorescence is detected during laser-off periods.

$$NO_2 + h\upsilon\ (\lambda = 449nm) \rightarrow NO_2{}^* \qquad\qquad\qquad\qquad R.\ 1$$

$$NO_2{}^* \rightarrow NO_2 + h\upsilon^{'}\ (\lambda \geq 449nm\ ) \qquad\qquad\qquad\qquad R.\ 2$$

The $NO_2$ fluorescence has a broad spectrum. It starts at the excitation wavelength and extends into the infra-red region (Wehry, 1976). But still, the major fraction of the fluorescence still lies in the visible region (Sakurai and Broida, 1969;Sugimoto et al., 1982). The detected fluorescence $h\upsilon^{'}$ is directly proportional to the amount of $NO_2$ in the cell. The background signal due to scattering and dark counts of the detector is determined by frequently measuring zero air (zero-$NO_2$). The atmospheric mixing ratios of $NO_2$ are derived by using Eq. 1.

$$NO_2 = \left[ \frac{Signal - S_{BG}}{\alpha_c} \right] \qquad\qquad\qquad\qquad Eq.\ 1$$

Where 'Signal' is in counts $s^{-1}$ and $\alpha_c$ is the calibration factor or sensitivity in counts $s^{-1}$ $ppb^{-1}$. $\alpha_c$ is derived from the slope of counts versus known amounts of $NO_2$. '$S_{BG}$' is the background signal in counts $s^{-1}$. The quality of zero air is further discussed in section 3.

The mechanical and optical parts of the LIF detection axis are presented schematically in Fig. 1. All mechanical parts inside GANDALF are black anodised and most optical components are continuously flush with zero air ($3 \times 50$ sccm) (Fig. 1, no.1) during the period of operation to avoid dead air pockets, fog, dust, etc. The inlet for the sampling flow line is a small orifice with a diameter of 0.7 mm. The distance from the point of entrance at the orifice to the centre of the detection cell (Fig. 1, no.2) is about 150 mm. This combination of orifice size and scroll pump provides a pressure of 7 hPa inside the detection cell, with a sampling flow of about 4100 sccm. The time required for air molecules from the point of entrance to reach the centre of the detection cell is less than 30 ms. The **diode laser**[2] (Fig. 1, no.3) in this system has a maximum output power of 250 mW with an on-off modulation frequency of 5 MHz. The wavelength ($\lambda$) of the diode laser is in the range of $447 \rightarrow 450$ nm. The

---

[2] Omicron Laserage (CW Diode-Laser), laserproduckte GmbH, Germany
  Power stability <1 % $hour^{-1}$, pointing stability: <10μrad
  Beam diameter: 2.55 (perpendicular: $0^{°}$/mm) & 2.53 (parallel: $90^{°}$/mm)

convolution of the laser profile and the $NO_2$ absorption cross-section (Vandaele et al., 2002) yields an effective $NO_2$ absorption cross-section of $5.3 \times 10^{-19}$ cm$^2$ molecule$^{-1}$. The laser beam is directed into the detection cell by using motorised mirrors (Fig. 1,

no.4). These mirrors are coated to achieve high reflectivity (99.8 %) for a light incidence at 45° with a wavelength of 450 nm. A **Herriot cell** (Herriott et al., 1964) is used to produce multiple passes to enhance the laser light, and focus at the centre of the detection cell. The detection cell of GANDALF is positioned between the Herriott cell mirrors (Fig. 1, no.5), which have approximately 99.99 % reflectivity (IBS coating)[3] in the spectral range of 445 nm to 455 nm. The distance between the mirrors is twice their radius of curvature (R = 128 mm). Any fluorescent contaminants from the mirrors are measured as a part of the

background signal. The multi-passed laser beam encompass a circle of about 8 - 10 mm diameter at the centre of the detection cell. A **photon counting head**[4] (PMT) is used for the fluorescence detection. The PMT is located in a tube (Fig. 1, no.6) perpendicular to the sampling flow line. The effective sensor area of the PMT is 5 mm in diameter and has a GaAsP / GaAs photocathode[5]. The PMT is sensitive to wavelengths between 380 nm and 890 nm, with peak sensitivity at 800 nm with a maximum quantum efficiency of 12 %. The fluorescence signal is focused onto the PMT by collimating lenses (Fig. 1, no. 7). An

aluminium concave mirror (Fig. 1, no.8) located opposite of the PMT redirects additional fluorescence photons towards the detector. In front of the PMT, **interference filters**[6] (Fig. 1, no.9) are used to remove contributions of light scattered from the walls of the sampling chamber, as well as from Rayleigh and Raman scattering. The filters have the cut-off wavelength (block radiation below this wavelength) of 470 nm and 550 nm respectively, with an average transmission of 98 % in the spectral range from cut-off wavelength + 3 nm to 900 nm. The reflectivity of the filters is higher than 99.7 % for the spectral ranges of about

8 nm below the cut-off wavelengths. The filters have a very small (< <1 %) absorption for almost the entire spectral regimes. However on the edge of the photonic stop band (cut-off wavelength) the absorption can be up to 7 % and 4 % for the filters with cut-off wavelength of 470 nm and 550 nm, respectively. At this positon, the photon density reaches to its maximum which increases the probability of absorption of a photon. If this absorption at about the cut-off wavelength exist than this can potentially amplify the luminescence. The fluorescence contamination is corrected using the background signal measurements.

An optical system (Fig. 1, no.10) based on photodiodes and a $NO_2$ filled cuvette is installed to monitor the change in the wavelength and power of the diode laser. The stray light in the system is reduced to a minimum by using a combination of baffles. There are different types of baffles (Fig. 1, no.11 and 12) used in the system to reduce scatter from walls or mirrors. The shape of a baffle surface is based on a zigzag pattern with a 30° angle. The sharpened edges of a baffle provide less surface area for the laser light to scatter and have the characteristics of a light trap.

The diode laser has a 'Deepstar' mode, which is used as an advantage for the system. While operating in this mode with the repetition rate of 5 MHz, there is no laser radiation during the off period and the $NO_2$ fluorescence is detected during the laser off period. To determine the optimum sensitivity as a function of the repetition range, the relative $NO_2$ fluorescence intensities for different on-off cycles has been calculated by taking into account key parameters like $NO_2$ absorption cross-section, pressure, flow velocity, fluorescence lifetime, and the power of the diode laser. The calculated sensitivity for different laser on-durations is

shown in Fig. 2 (Left-side) based on 1 ppb of $NO_2$ as a function of off-period duration. For a comparison to current operational on-off cycles, three different on-periods are shown in Fig. 2 (Left-side). The best sensitivity of the instrument is achievable by operating the diode laser at 5 MHz, 100 ns on, 100 ns off.

A counter card is used for the data acquisition. There is no need for synchronisation as the counter card itself triggers the laser pulse. The timing system is entirely controlled by an FPGA (field-programmable gate array), utilizing an external

---

[3] ATFilms (IBS coating), USA
[4] Hamamatsu (H7421-50), Japan, Count sensitivity: $2.1 \times 10^5$ s$^{-1}$pW$^{-1}$ at 550 nm and $3.9 \times 10^5$ s$^{-1}$pW$^{-1}$ at 800 nm
[5] Radiant sensitivity of 87.4 mA W$^{-1}$
[6] Barr Associates, Inc., USA

crystal oscillator of 20MHz nominal frequency with a stability of +/-2.5ppm over the temperature range of -30°C to +75°C. All internal frequencies are derived from this clock by means of a PLL (phase-locked loop) in the FPGA. The triggering occurs at a fixed rate of 5 Mhz. The delay caused by the length of the trigger cable (propagation delay of the pulse), the laser power supply unit, propagation delays from detector to FPGA, etc. is compensated with a programmable delay for the data acquisition in the FPGA. So the FPGA logic recognises when it should start recording the data after it emitted the trigger pulse and waits the

specified amount of programmed clock cycles after emitting the trigger. The time-resolved raw signal (both on-off cycle) are stored in 4 ns bins (4 ns bin = 1 channel) for a specified time of integration (typically 1 s). For the total fluorescence signal about 20 of these channels are summed up and used as a signal for $NO_2$. The first 5 channels or 20 ns of the laser off period are ignored because these channels still contain some scattered light signals from the laser light and walls of detection cell [Fig. 2 (Right-side)].

The surface temperature of the PMT and laser is kept at a temperature of 20°C or 25°C (avoiding condensation) by circulating water. This provides the heat sink for the internal thermoelectric cooling of the PMT (@ 0°C) and laser (@ 25°C). The internal cooling is the default setting from the manufacturer. The internal temperature cannot be regulated by an external cooling. The external temperature should be in the range of 5-35°C along a sufficient heat exchange system (fan cooling, water circulation etc.). Moreover, the dark counts on the PMT signal are in the order of < 50 counts $s^{-1}$ for the channels used for the

$NO_2$ fluorescence detection. The major reason for the background signal, larger than the dark signal typically by a factor >25, is expected to be fluorescence contamination from the Herriot cell mirrors existing in the red region of wavelength. For a stable (parameters like power, wavelength, shape of the beam etc.) laser operation, an external temperature range is within 15-30°C. This range is sufficient to keep the internal temperature of the laser at 25°C. A laser operation out of the specified range would lead to shut-off/potentially damage the laser.

## 2.2 Calibration system

The LIF method is not an absolute technique and requires calibration. The sensitivity (Eq. 1) of GANDALF depends on e.g. background noise, laser power or wavelength, temperature, pressure, residence time in the sampling line, etc. It is determined using $NO_2$ concentrations generated by gas phase titration of NO to $NO_2$ by means of $O_3$ (R. 3) similar to the one described by

e.g., (Ryerson et al., 2000). Using commercial available $NO_2$ gas cylinders at low concentrations (Thornton et al., 2000;Dari-Salisburgo et al., 2009), was not chosen due to its open questions with its long term stability at low concentration. The calibration system is described in the following sections.

        The NO calibration mixture for the gas phase titration is traceable to a primary NIST[7] standard (4.91 ± 0.04 μmol mol$^{-1}$ in nitrogen). The overall uncertainty of the NO calibration mixture is 2 %. NO is almost completely (> 98 %) consumed during

gas phase titration with $O_3$ in the calibrator. This is achieved by using a high concentration (> 1.4 ppm) of $O_3$. $NO_2$ also reacts with $O_3$ to form $NO_3$ (R. 4). The reaction of $NO_2$ with $O_3$ (R. 4) is slower by 3 orders of magnitude compared to the reaction of NO with $O_3$ (R. 3), with a reaction rate of $3.5\times10^{-17}$ cm$^3$ molec$^{-1}$ s$^{-1}$ compared to $1.8\times10^{-14}$ cm$^3$ molec$^{-1}$ s$^{-1}$ at 298 K (Atkinson et al., 2004), respectively. However, at higher concentrations and due to the long residence times, the reaction between $NO_2$ and $O_3$ can become important, leading to a loss of $NO_2$ generated in the calibrator with subsequent losses due to further reaction between

$NO_2$ and $NO_3$ (R. 5).

$NO + O_3 \rightarrow NO_2 + O_2$                               R. 3

---

[7] National Institute of Standards and Technology, USA

$$NO_2 + O_3 \rightarrow NO_3 + O_2 \hspace{8cm} \text{R. 4}$$

$$NO_2 + NO_3 + M \leftrightarrow N_2O_5 + M \hspace{7cm} \text{R. 5}$$

Numerical simulations are used to assess the optimum setup for the calibration device by studying the impact of different parameters like concentrations levels, residence time, flow rates, pressure, etc. Based on box model (BM) simulations and

195 verified by lab experiments, a PFA (Perfluoroalkoxy) reaction chamber for the completion of the gas phase titration between NO and $O_3$ has been designed to achieve maximum conversion efficiency for $NO \rightarrow NO_2$. The BM simulation is shown in Fig. 3 for typical calibration parameters[8]. The mixing ratios of NO, $O_3$, and $NO_2$ are plotted as a function of residence time in the left panel of Fig. 3. This simulation predicts that more than 99 % of NO is converted to $NO_2$ within the residence time of 7.5 s inside the reaction chamber. The formation of $NO_3$ and $N_2O_5$ in the reaction chamber is negligible ($< 0.5$ ppb) compared to $NO_2$

($> 100$ ppb). The formation of $NO_3$ and $N_2O_5$ can thus only raise $< 1$ % uncertainty in the generated $NO_2$ for typical operating conditions of the calibrator. After the reaction chamber, the calibration gas mixture is further diluted with zero air to achieve a required range (close to ambient levels) of $NO_2$ mixing ratios.

The calibration system was tested for different concentrations of $O_3$. Figure 4 shows the $NO_2$ signal of the PMT (after dilution of the calibration gas) based on different $O_3$ mixing ratios in the reaction chamber for a constant NO concentration

(about 0.1 ppm). For $O_3$ concentrations below 1 ppm non stoichiometric conversion of NO was observed as expected. The PMT signal reached a maximum at about 1.356 ppm and this signal is explained by the derived $NO_2$ concentration from the BM simulation. The decrease of the PMT signals at higher $O_3$ concentrations above 1.4 ppm mainly due to loss of $NO_2$ in reactions R. 4 and R. 5. The amount of $NO_2$ generated in the 'NO + $O_3$' titration is much less sensitive to $O_3$ than to NO as losses of $NO_2$ (R. 4) also being dependent on $O_3$ for the chosen parameters. If $O_3$ is increased by 1 ppm above the optimum mixing ratio of

1.3 ppm (Fig. 4), $NO_2$ is reduced by only 1 %. The $O_3$ concentrations are always kept above this threshold limit and the concentrations are measured using an $O_3$ analyser[9] with a typical precision of 5 %. Above the threshold, a 5 % change in $O_3$ produces an uncertainty in $NO_2$ of less than 0.5 %.

A $NO_x$ analyser[10] was used to determine the remaining concentrations of NO inside the calibrator after the gas phase titration. About 99 % of NO is consumed in the gas phase titration for most of the cases at $O_3 > 1.4$ ppm. There are two different

regimes in the calibration system based on different NO and $O_3$ concentrations and different flow rates: (1) gas phase titration in the reaction chamber and (2) dilution with zero air after the reaction chamber. Considering the flow rates and dimensions of the gas lines, the theoretically calculated total residence time based on the plug flow assumption is 7.73 s. While the total residence time for the calibration gas in the calibration system is also determined experimentally by using Eq. 3.

$\hspace{1cm} NO_{GPT} = D[NO_i]\left(e^{-k_{R.3}[O_3]t_1}\right)\left(e^{-k_{R.3}[O_3]Dt_2}\right) \hspace{6cm} \text{Eq. 2}$

---

[8] For this specific simulation, initial parameters;
      NO = 5 sccm × 10.55 ppm,
      $O_3$ = 500 sccm × 1.7 ppm,
      residence time in the reaction chamber = 7.5s,
      flow = 8000 sccm
      temperature and pressure = 298 K and 1013.25 hPa
[9] ANSYCO, O3-41M, 'Analytische Systeme und Componenten GmbH', Germany
[10] Model: ECO PHYSICS CLD 780 TR, Switzerland

$$\Rightarrow [t_1 + Dt_2] = -\left[\frac{ln\left(\frac{NO_{GPT}}{D[NO_i]}\right)}{k_{R.3}[O_3]}\right] \qquad \text{Eq. 3}$$

In Eq. 3, '$NO_{GPT}$' is measured with the $NO_x$ analyser and is defined as the NO concentration remaining in the calibration gas
after the gas phase titration and dilution. $[NO_i]$ is the initial concentration of NO before the gas phase titration. 'D' is the dilution factor after the reaction chamber. '$t_1$' is the residence time for the reaction chamber and '$t_2$' is the dilution dependent travel time for a $NO_2$ molecule from the exit of the reaction chamber to the inlet of GANDALF. $[O_3]$ in Eq. 3 is the concentration in the reaction chamber. '$k_{R.3}$' is the temperature dependent rate coefficient for R. 3. There are two slightly different (< 6% based on rate constant at 298K) values reported in the literature for the temperature dependent $k_{R.3}$ as follows:

$$k_{R.3} = 3 \times 10^{-12} \times e^{\left(\frac{-1500}{T}\right)} \qquad (1.9 \times 10^{-14} cm^3 molec^{-1}s^{-1} \text{ at } 298K) \qquad \text{(Sander et al., 2011)}$$

$$k_{R.3} = 1.4 \times 10^{-12} \times e^{\left(\frac{-1310}{T}\right)} \qquad (1.8 \times 10^{-14} cm^3 molec^{-1}s^{-1} \text{ at } 298K) \qquad \text{(Atkinson et al., 2004)}$$

Based on Eq. 3, the average value of total residence time $[t_1 + D\,t_2]$ is 7.32 s ± 0.25 s (Sander et al., 2011) or 8.38 s ± 0.29 s (Atkinson et al., 2004) as shown in Fig. 5. The estimated accuracy of these two values for the total residence time is 6.5 % (1σ).

The temperature and pressure also affect the formation of $NO_2$ inside the reaction chamber, and these effects were tested with the box model. In the simulations all parameters except temperature or pressure are kept constant. At a lower temperature the reaction between NO and $O_3$ slows down leading to changes in the conversion efficiency from NO to $NO_2$. This can
potentially lead to a change in the conversion efficiency from NO to $NO_2$. In our case, many electrical parts (electronic valves, ozone generator, and mass flow controllers) are installed inside the calibration unit. In a fully operational mode for one day, the temperature build up in the calibration unit is 8-10°C higher than ambient temperatures. From our experience/observations, conditions with a temperature lower than 20°C inside the calibrator do not occur. According to the box model simulations temperature variations within 5 - 45 °C leads to an overall relative uncertainty of 1 % (1σ) for the whole range. Similarly, the
impact on the calibration gas due to a change in the atmospheric pressure is not significant. Based on the box model simulations, the relative uncertainty in the $NO_2$ concentration of calibration gas due to a change in the atmospheric pressure over an interval of 800 – 1013 hPa is below 0.5 % (1σ).

The calibration gas for GANDALF primarily contains $N_2$ (~79.5 %) and $O_2$ (~20.5 %) with $H_2O$ vapour (< 25 ppm). The level of $H_2O$ vapour in the atmosphere can reach up to about 3 % (Seinfeld and Pandis, 2006). The sensitivity of
the instrument is reduced by atmospheric $H_2O$ vapour because collisions with $H_2O$ molecules quench the $NO_2$ fluorescence. The $H_2O$ dependency is evaluated experimentally by diluting the calibration gas with known amounts of water vapour concentrations and its effect on sensitivity during field measurements is corrected by using simultaneous measurement of $H_2O$ vapour in the atmosphere. The $H_2O$ concentrations during calibration are determined using an existing calibration system for the LIF-OH instrument (Martinez et al., 2010). The decrease (relative to < 25 ppm of water vapour) in the sensitivity for GANDALF is
5 % ± 1 % (1σ) at 1 % of atmospheric $H_2O$ vapour.

A robust calibration system has been developed for the automated calibration of the instrument. GANDALF is frequently calibrated (up to 8 times in 24h) during field operations to track changes in sensitivity. Generally, some factors can contribute to a change in the sensitivity e.g., stability of the optics alignment, cleanness of the optics, temperature related effect of electronics, stability of the calibration signal etc. An example for a calibration plot is shown in Fig. 6. The calibration system

is controlled by Mass Flow Controllers (MFC)[11] and electronic valves[12]. All MFC are calibrated using a DryCal[13] sensor which is traceable to a NIST standard (NIST traceability is confirmed by Westphal[14]). The uncertainty in the set flows, based on a certified value, is 1 % (level of confidence 95 %). $O_3$ is generated for the calibration using an ozone generator[15]. Different $NO_2$ mixing ratios are achieved by changing the NO flow (range up to 10 sccm), while the $O_3$ concentration (> 1.4 ppm) and flow (500 sccm) are kept constant. Figure 7 shows a schematic of the setup for the automated calibration procedure of GANDALF. A

small pump (calibration pump) is connected to the main sampling line of GANDALF. A three-way electronic valve (EV2) and a manual needle valve (MNV) are attached in front of the calibration pump. To minimise any line effects such as a decomposition of species like PAN, the chemical reaction of the ambient NO and $O_3$, etc., the residence time in the sampling line is kept at less than 0.1 s by a flow $\geq$ 10000 sccm required during ambient air measurements. GANDALF has a flow of 4100 sccm through the pin hole and the rest of the flow is diverted to the main exhaust by the calibration pump. The amount of total sampling flow can

be increased or decreased by adjusting the manual valve MNV.

        During ambient air measurements, valve EV2 is opened for line L1 at the position P1 (Fig. 7) and allows an extra flow of about 8000 - 9000 sccm to pass from the sampling flow to the calibration pump. Line L1 is simultaneously used to condition the NO calibration line with a flow of 2 sccm NO gas, which goes directly to the exhaust without entering the sampling line. The direction for the conditioning flow along the bypass flow is shown by the green arrow in Fig. 7.

Frequent zero-air measurements are necessary to monitor changes in the background signal of GANDALF. A three-way electronic valve (EV3) and a mass flow controller (MFC Zero) are used to switch the zero air background flow (8000 sccm) on and off in the line L3 (position P1 at EV3 in Fig. 7). During background signal measurements, an excess of zero-air about 3900 sccm (blue arrow in Fig. 7) is diverted to the calibration pump through line L1 by setting the valve EV2 to position P1, along with about 5100 sccm flow of ambient air.

During calibration the zero air flow is switched on (position P2 at EV3 in Fig. 7) and used for dilution of the calibration gas. Line L2 is opened by valve EV2 (position P2 at EV2 in Fig. 7) to remove the calibration gas overflow of 3900 sccm together with 5100 sccm from the ambient (illustrated by the red arrows in Fig. 7). For the gas phase titration, the flow of $O_3$ is switched on and off by the two-way electronic valve EV1 and MFC ($O_3$). The $O_3$ analyser is used to check the concentration of $O_3$ produced by the ozone generator. The flow of NO (1 - 10 sccm) is controlled by a mass flow controller [MFC (NO) in Fig. 7].

Since all overflows are diverted to an exhaust, this setup allows frequent checks of the GANDALF sensitivity and background signal without disturbing the ambient conditions for a nearby operating instrument. Based on calibrations during PARADE-2011, the repeatability of the sensitivity was 2.7 % (1$\sigma$), with an overall uncertainty of the calibration system of approx. 5 % (1$\sigma$).

## 2.3 Precision and limit of detection

The precision of the instrument was evaluated using a set of randomly chosen PMT signal (in s[-1] time resolution) during calibration periods from the field experiment (PARADE-2011). The relative precision was calculated based on the standard deviation of the PMT $NO_2$-signal for different $NO_2$ concentrations. The relative precision of GANDALF is shown in Fig. 8 as a function of $NO_2$ mixing ratios. It was better than 0.5 % (1 $\sigma$ min[-1]) for most of the dataset at >1ppb of $NO_2$. For an overall precision of GANDALF (especially at lower levels < 1ppb), an absolute value of about 3 ppt (1$\sigma$) has to be added to the relative

---

[11] MKS Instruments and Bronkhorst HIGH-TECH B.V, USA
[12] Solenoid Operated Diaphragm, Galtek, USA
[13] DC-2, BIOS International Corporation , USA
[14] WESTPHAL measurement and control technique GmbH & Co. KG, Germany
[15] SOG2, 185nm, UVP - Ultraviolet Products, USA

precision. This absolute value arises from the variations in the zero-air signal. The standard deviation of the PMT signals at different $NO_2$ concentrations can be extrapolated to zero for determination of the precision at background levels. It can also be calculated from the standard deviation of the zero-air signal. Both approaches give a similar result of about 3 ppt (1σ) precision for our instrument.

The precision of the instrument background signal was also cross-checked using a continuous measurement of zero-air for about 50 minutes. In order to verify the square root dependency of the signal variability on integration time, an Allan deviation plot is used (Riley, 1995; Land et al., 2007). Figure 9 shows an overlapping (Riley, 2008) Allan deviation plot of variations in background signal versus averaging time. The variations in background signal with a 60 s integration time are equivalent to an absolute $NO_2$ value of about 3 ppt (1σ). Figure 9 also shows that the random noise of the instrument background signal can be reduced by averaging, with a square root dependency on time, at least up to a 60 s period. The background signal of GANDALF is frequently checked during a field operation (e.g. during PARADE, 1 background signal measurement per hour).

The limit of detection (LOD) can be derived from the variation of the background signals. Based on the Allan deviation plot in Fig. 9, a limit of detection of about 3 ppt (1σ) $NO_2$ for one minute averaged measurements is expected. The stated (Table 1) LOD of GANDALF was calculated using Eq. 4 (Taketani et al., 2007) at a signal-to-noise ratio SNR of 2 and considering the two times higher background signal.

$$LOD = \frac{SNR}{\alpha_c} \sqrt{\frac{2 \times S_{BG}}{t}}$$ 
Eq. 4

Where $\alpha_c$ is the calibration factor or sensitivity in counts ($s^{-1}$ $ppb^{-1}$), $S_{BG}$ is the background signal in counts ($s^{-1}$) and t is the averaging time in seconds. The LOD for GANDALF, based on sensitivity and background measurements during the field experiment (PARADE-2011), varied between 5 and 10 ppt.

## 2.3 Interferences by other species

Several atmospheric gas species can absorb the 449 nm laser light inside the detection cell. This can lead to interference for the $NO_2$ measurements with GANDALF directly (photodissociation process) or indirectly (fluorescence).

Iodine monoxide (IO) has an absorption cross-section of $3.9 \times 10^{-18}$ $cm^2$ $molecule^{-1}$ (Harwood et al., 1997) and is about a factor 8 larger than the $NO_2$ absorption cross-section at 449 nm. Even a few ppt of IO in the atmosphere can produce a significant fluorescence signal, especially in the marine atmosphere for which IO is mostly reported (Commane et al., 2011) . The fluorescence lifetime of IO is only 1-10 ns (Bekooy et al., 1983; Newman et al., 1998). As described earlier, the initial 20 ns fluorescence signal is ignored in the GANDALF data evaluation. So the IO fluorescence signal after 20 ns becomes too small to significantly interfere with the $NO_2$ fluorescence signal.

Nitrogen containing inorganic species ($NO_3$, $N_2O_5$, $HONO_2$, $HO_2NO_2$, PAN, ClONO, $ClNO_2$, and $ClONO_2$) can produce $NO_2$ by photodissociation which can happen inside the detection cell. $N_2O_5$ (Harwood et al., 1993), $HONO_2$ (Burkholder et al., 1993), $HO_2NO_2$ (Singer et al., 1989), PAN (Talukdar et al., 1995), and ClONO (Molina and Molina, 1977) are not known to photo-dissociate at this wavelength. The absorption cross-sections for $ClONO_2$ (Molina and Molina, 1979) and $ClNO_2$ (Ghosh et al., 2012) are smaller by about 4 orders of magnitude compared to that of $NO_2$ at 449 nm. The tropospheric concentrations of $ClONO_2$ and $ClNO_2$ are generally smaller to similar compared to ambient $NO_2$. Hence, an interference from these species is highly unlikely.

NO$_3$ has a larger absorption cross-section (Wayne et al., 1991) at 449 nm compared to the previously described nitrogen-containing species. The effective absorption cross-section, calculated from (Wayne et al., 1991), is about a factor of 2 smaller than that of NO$_2$ at the wavelength of the diode laser. The recommended quantum yield for the photodissociation of NO$_3$ to NO$_2$ + O is about 1 at wavelengths below 585 nm (Sander et al., 2011); hence, its fluorescence (Wood et al 2003) is insignificant compared to its photodissociation to NO$_2$. Interference from photodissociation of NO$_3$ is therefore a two-photon process:

1$^{st}$ step: NO$_3$ + h$\upsilon_{DiodeLaser}$ → O + NO$_2$        2$^{nd}$ step: NO$_2$ + h$\upsilon_{DiodeLaser}$ → NO$_2$$^*$ → NO$_2$ + h$\upsilon$        R. 6

The lifetime of NO$_3$ can be estimated from Eq. 5.

$$\tau(NO_3) \approx \int \sigma_{NO_3}(\lambda, T) \times \varphi_{NO_3}(\lambda, T) \times F(\lambda, T)\, d\lambda \qquad \text{Eq. 5}$$

Where σ (λ, T) is the effective absorption cross-section of NO$_3$ which is 2.7×10$^{-19}$ cm$^2$ molecule$^{-1}$; φ (λ, T) is the quantum yield for NO$_3$, and F (λ, t) is the photon flux from the diode laser of about 10$^{20}$ photons cm$^{-2}$ s$^{-1}$. The residence time of sampling air in the effective beam area of the laser is much smaller (<0.001 s) compared to the NO$_3$ photodissociation lifetime (> 0.01 s). Due to this reason, any chance of a significant interference from the NO$_3$ photodissociation is highly unlikely. Moreover, the ratio of the atmospheric concentration between NO$_2$ and NO$_3$ is very high, e.g. during PARADE the median ratio NO$_2$ / NO$_3$ was 430 for NO$_3$ > 0 with a minimum value of 12.

Alkenes and aromatics (aldehydes and benzene) are also abundant in the troposphere. However, absorption of alkenes and aromatics occurs in the UV range (< 300 nm) (Keller-Rudek et al., 2013), well below the wavelengths used in GANDALF. Some carbonyls like glyoxal (CHOCHO), and methylglyoxal (CH$_3$COCOH) also have absorption in the blue region of the visible spectrum. The absorption cross-section values of CHOCHO, and CH3COCOH are 5.28 ×10$^{-20}$ cm$^2$ molecule$^{-1}$ (Horowitz et al., 2001), and 9.26×10$^{-20}$ cm$^2$ molecule$^{-1}$ (Meller et al., 1991;Staffelbach et al., 1995) at 449 nm, about a factor 10 and 5 smaller than the NO$_2$ absorption cross-section, respectively. Also the fluorescence from these species is not known to be present in the region of NO$_2$ fluorescence. So, the interference from these species is not important.

To minimise the impact (prior to the orifice) of heterogeneous or thermal conversion of species like PAN (*lifetime*[16] ≈ 2327 s), HO$_2$NO$_2$ (*lifetime* ≈ 16 s), CH$_3$OONO$_2$ (*lifetime* ≈ 0.3 s), and N$_2$O$_5$ (*lifetime* ≈ 22 s) yielding NO$_2$, a short residence time of < 0.1 s is generally used by keeping the sampling flow high, e.g. 12000 sccm in a 0.5 m long sampling line with a 4 mm internal diameter during PARADE-2011. After the orifice, the cell pressure is about 7 hPa and this would lead to increase even further the lifetime of above-specified species. Whereas the residence time after the orifice is less than 30 ms. So a chance of interference in the low-pressure region from the thermal conversion is highly unlikely. An intercomparison of GANDALF and other measurements of NO$_2$ during PARADE-2011 was conducted to look for systematic dependencies of the differences between the different measurements on several measured atmospheric quantities. No evidence for a potential interference has been found for GANDALF (Section 3.2).

---

[16] The *lifetime* is calculated from IUPAC rate coefficient (temperature = 298 K and pressure = 1 bar) for the sampling line before the orifice.

## 3 Field Experiment: PARADE-2011

The PARADE, **PA**rticles and **RA**dicals: **D**iel observations of the impact of urban and biogenic **E**missions, field experiment took place at the Taunus Observatory on Kleiner Feldberg (825m ASL[17]; 50.22° N, 8.45° E) in Germany from the 15th of August (DOY[18] = 226) to the 10th of September (DOY = 252) 2011. The general focus of PARADE was to characterise summertime biogenic emissions and photochemistry in a forested environment with anthropogenic influence. The observatory is located in the vicinity of the Taunus region at the hilltop of Kleiner Feldberg. A total area of 5 km radius around the observatory is

dominated by coniferous, broad leaved and mixed forest. The measurement platform was located at the top of the observatory. The site is often affected by anthropogenically influenced air from nearby cities such as Frankfurt/Main (30 km SE), Wiesbaden (20 km SW), Mainz (25 km SSW), and some roads within 5 - 10 km, depending on the wind direction. The temperature during PARADE varied within a range of 5 - 28 °C with an overall average of 14.8°C. The temperature conditions during PARADE can be subdivided into two phases. The periods of DOY = 226 - 237 and DOY = 243 - 246 for PARADE were slightly warmer and

the temperature mostly stayed above 15 °C, whereas during the other periods of DOY = 238 - 242 and DOY = 248 - 252 the temperature was below 15°C. The relative humidity (RH) had an overall average value of 77 % and variations within the interval of 37-100 %. There were several episodes of rain during PARADE. In the later part of the campaign, fog persisted in the early morning hours. Air masses at the observatory arrived predominantly from the southwest (SW) to the northwest (NW) side of Kleiner Feldberg. Sampling lines for most of the trace gas monitoring instruments were located within a 5 $m^2$ area at the top of

the platform. The platform was about 8 m above ground and the top of the platform was above the forest canopy. An overview of the instrumentation and conditions during PARADE can be found e.g. in (Phillips et al., 2012;Bonn et al., 2014). Note that all data sets for analysis are based on available 10-minute averages.

### 3.1 $NO_2$ inter-comparison during PARADE

$NO_2$ concentrations were measured with eight different instruments. Six out of eight instruments sampled at the top of the platform. The measurement techniques, uncertainties, time resolutions and LOD are summarised in Table 2 for the instruments located on the platform. The average ambient concentrations of $NO_2$ during PARADE were approx. 2 - 3 ppb with a range of approx. 0.13 ppb to 22 ppb. $NO_2$ instruments listed in Table 2 represent in situ measurement techniques with the exception of the LP-DOAS (Long Path Differential Optical Absorption Spectroscopy).

A median value (based on 10 minute averages) of the atmospheric $NO_2$ concentration is derived from the $NO_2$ measurements of all individual instruments at the platform including LP-DOAS. For a valid correlation between the derived median $NO_2$ and individual $NO_2$ measurements, only those values of the median $NO_2$ were selected for which simultaneous data for all $NO_2$ measurements were available. Figure 10 shows the correlation between individual $NO_2$ measurements and the derived median $NO_2$ concentrations. The total uncertainties of individual instruments are shown as error bars on the y-axis while

horizontal bars represents the standard deviation of the derived median $NO_2$. The regression is done by using a 'bivariate' fit according to the method described in (York et al., 2004;Cantrell, 2008).

     **LP-DOAS:** This instrument is based on traditional Differential Optical Absorption Spectroscopy (DOAS) (Platt et al., 1979;Perner and Platt, 1979), and follows the Beer-Lambert law. DOAS allows direct and absolute measurements of multiple trace gases in the atmosphere by using the distinct absorption band structure of the specific molecule (i.e. calibration is not

needed) (Platt and Stutz, 2008). LP-DOAS is based on active remote sensing and requires an artificial light source (Pöhler et al.,

---

[17] above sea level
[18] DOY (Day of year 2011)

2010). It provides an average concentration of $NO_2$ or other trace gases through quantitative detection using the absorption over a light path of typically a few kilometres. The instrument in this study is a well-established instrument and has been a part of many field campaigns (Pöhler et al., 2010). During PARADE, the optical path length was approximately 2.5 km and the light source as well as the spectrograph was located on the platform. The optical retro-reflector reflecting the light back to the telescope was
located on the mountain Großer Feldberg (Distance =1.23 km and Height = 37 m).   Therefore the LP-DOAS measurement delivers values integrated along a 1.2 km straight line starting at the platform to the retro-reflector. The correlation ($R^2 = 0.96$) plot between LP-DOAS and the derived median $NO_2$ values is shown in subplot E of Fig. 10. The slope of the fit is $1.02 \pm 0.005$ with a negligible y-intercept of $-0.002 \pm 0.009$ ppb and these values are within the uncertainty of the instrument. The uncertainty of LP-DOAS is mainly due to errors in the absorption cross-sections of $NO_2$. A larger scatter between the LP-DOAS
to the in situ instruments is expected due to the sampling of different air masses (A1 in Fig. 11).

**CE-DOAS:** Cavity-Enhanced DOAS (Platt et al., 2009) measurements of $NO_2$ were also available during PARADE. This method is based on Differential Optical Absorption Spectroscopy (DOAS) combined with a cavity and provides in situ measurements of trace gases (Platt et al., 2009). CE-DOAS requires calibration of the absorption light path in the cavity. This was performed with the measurement of two different Rayleigh absorbers (Helium, and air) according to (Washenfelder et al.,
2008). The campaign was also the first field trial for this instrument with a reported error of measurements in the range of 5 – 10 %, mainly due to the accuracy of the light path calibration. The CE-DOAS and the CRDS shared the same sampling line. The slope and the y-intercept for CE-DOAS versus the median derived $NO_2$ is $0.92 \pm 0.007$ and $-0.032 \pm 0.01$ ppb, respectively, with $R^2 = 1$ as shown in subplot [F] of Fig. 10. The difference to the median value is well within the range of instrumental uncertainty of this prototype. A further development of this prototype is the ICAD (iterative cavity enhanced DOAS) from Airyx
GmbH.

**CRDS:** Besides the DOAS instruments, another $NO_2$ measurement technique using a Cavity Ring-Down Spectrometer (CRDS) was available (Thieser et al., 2016). CRDS is a cavity-assisted method like CE-DOAS (Platt et al., 2009). It is a direct method for in situ measurements which requires no calibration but only the background (zero-air) measurements.  In CRDS, reflective mirrors are used across an optical cavity. To obtain the concentration of a trace gas with CRDS, absorption
measurements to determine the time constant for exponential decay of the light intensity with and without an absorber are made in the optical cavity. During PARADE, the instrument inlet was located 2 m above the platform. An about 8 m long PFA tube was used for the sampling air. The slope and y-intercept in the case of CRDS are $1.06 \pm 0.007$ and $0.01 \pm 0.01$ ppb with correlation $R^2 = 0.99$ as shown in panel D of Fig. 10. The reported upper limit of uncertainty in the case of CRDS is $[6 \% + 20 \text{ ppt} + (20 \text{ ppt} \times RH^{19}/100)]$ (Thieser et al., 2016). The differences between CRDS and the derived median $NO_2$ values
are smaller than the instrument errors.

**CLD/Blue-light converter (BLC):** Along with the above mentioned absolute methods, the concentrations of $NO_2$ and NO were determined with a two-channel chemiluminescence detector (CLD). The instrument sampled air via ~8 m long PFA tubing at 2 m above the platform. The CLD instrument of MPIC is well-established, being an improved version (Hosaynali Beygi et al., 2011) of the ECO-Physics CLD 790 SR. In this instrument, $NO_2$ is detected by conversion via photolysis to NO,
using a blue light converter at the wavelength of 395 nm, with subsequent detection of NO by chemiluminescence. The calibration of the system is done by using gas phase titration between NO and $O_3$ to produce stoichiometric quantities of $NO_2$. The correlation ($R^2 = 0.99$) between CLD and the derived median $NO_2$ values is shown in panel C of Fig. 10. Overall, the data of the CLD is about 5 % below the median, but this difference is within the uncertainty of the CLD measurement. The reported uncertainty of the CLD for the $NO_2$ measurements is 105 ppt or 10 % (Li et al., 2015). The slope and y-intercept are

---

[19] Relative humidity in %

$0.95 \pm 0.008$ and $-0.1 \pm 0.01$ ppb, respectively. A larger negative intercept could be related to measurements of higher background for the BLC unit (switch ON) leading to underestimation of ambient $NO_2$. An additional background signal is most likely due to decomposition of surface absorbed NO or $NO_2$ during the operational mode of the BLC unit (Teflon block).

**GANDALF:** The sampling flow rate (12000 sccm) provided a residence time of less than 0.1 s in a 0.5 m sampling line. This was sufficient to suppress the impact of heterogeneous or thermal conversion of $NO_2$ containing species to yield $NO_2$. The

formation of $NO_2$ due to the reaction between ambient NO and $O_3$ in the sampling line was negligible. The campaign averages of the observed concentrations of NO, $O_3$ and $NO_2$ were 0.25 ppb, 44 ppb and 2.6 ppb respectively. Based on average NO and $O_3$ concentrations, the formation of $NO_2$ from the reaction 'NO + $O_3$' in the sampling line was less than 0.01 % with respect to the ambient $NO_2$ concentrations. Line loss or photolysis of $NO_2$ was avoided by using PTFE lines (Polytetrafluoroethylene) covered with a dark insulating material. The average pressure inside the detection cell for the entire period of PARADE was

$6.95 \pm 0.27$ (1σ) hPa. Several automated calibrations (2 - 8 per day) and background level measurements (once per hour) were conducted during PARADE to ensure the precision and accuracy of the instrument. Based on the hourly background level measurements, we established that the deviations for about 70 % (1 σ) of successive background signal measurements (no. of measurements > 500) were within an equivalent value of ± 8 ppt of $NO_2$. Any $NO_2$ impurity in the used zero air[20] (Synthetic air hydrocarbon free, without subsequent scrubbing) would lead to under estimation of ambient $NO_2$ levels for PARADE and further

contribute to the uncertainty. Nevertheless, previously describe deviations of 8 ppt in the background signal during PARADE could be a good indicator for this uncertainty. Another indication that the $NO_2$ contamination in zero air used during PARADE-2011 was less than GANDALF's detection limit is that in the data analysis the y-intercept of other $NO_2$ in situ instruments (y-axis) vs GANDALF (x-axis) showed always a negative number. If the GANDALF zero-measurements would have significant $NO_2$ contamination the y-intercept should be positive (This figure is provided in the supplement). A malfunction of the $O_3$

generator occurred in the period 4 to 10 September that disturbed the GANDALF calibration system. A correction of 12 % is introduced for the period 4 − 10 September, based on the correlation of GANDALF with the CRDS instrument prior to 4 September. During the last few days of this period, an extra baffle was installed in GANDALF. The baffle can be inserted easily into the detection block of GANDALF without disturbing the alignment of the laser. The advantage of the baffle is that it reduces the background counts by ~50 % while decreasing sensitivity by less than 5 %. The overall correlation between

GANDALF and the derived median $NO_2$ is $R^2 = 0.99$ as shown in panel B of Fig. 10. The measurements of GANDALF tend to be 3 % higher compared to the derived median values of $NO_2$. This overestimation of slope from unity compared to the derived median value is within the range of the instrument uncertainties. The overall relative uncertainty of GANDALF during PARADE was about 5 % + 11 ppt and it showed an exponentially increasing trend from a higher to lower concentration of $NO_2$. This increasing trend is mainly driven by the error in the background measurements. The slope and y-intercept of the fit are 1.03 and

0.027 ppb with the absolute error of the fit being 0.006 and 0.01 ppb, respectively.

Generally, all instruments for $NO_2$ showed reasonable agreement with the derived median $NO_2$. Based on Fig. 10, GANDALF (+ 3 %), CRDS (+ 6 %) and LP-DOAS (+ 2 %) showed over-estimation compared to the derived median values while the data from CLD was about − 5 % and from CE-DOAS about − 8 % lower than the median values. The overall differences are within the experimental limitations and instrumental uncertainties. Results of the comparison between individual

$NO_2$ measurements and the derived median $NO_2$ at different ranges of $NO_2$ mixing ratios are summarised in Table 3.

---

[20] (Synthetische Luft, KW frei 12er MBdl) Westfalen AG, Germany

### 3.2 Ratio distribution of $NO_2$ measurements

Various measurements of trace gases, meteorological parameters, and photolysis frequencies during PARADE provided an opportunity to look for indications of systematic differences between $NO_2$ instruments. Ratios of the individual $NO_2$
485 measurements to GANDALF, which are referred to as "ratios" further in this section, are compared in respect of different atmospheric conditions. The distribution of ratios is shown as a histogram in the upper panel of Fig. 11 [A1, A2, A3, and A4] along their respective fits based on the normal distribution. The 'normal probability plot' for empirical probability versus ratios is shown in the lower panels [B1, B2, B3, and B4] of Fig. 11. This plot is a graphical representation of the normal distribution of ratios. The plot stays linear as long as the distributions are normal, and the deviation from the linear fit shows the divergence
490 from the normal distribution. The solid line in the lower panels of Fig. 11 is between the $25^{th}$ and $75^{th}$ interquartile range of a ratio. The probability's grid (y-grid lines) is not linear and it is representative of the distance between quantiles of normal distribution.

 The average, median, and standard deviations of ratios comparing GANDALF with other instruments are given in Table **4**. The variation in these ratios (CRDS/GANDALF, CE-DOAS/GANDALF, and CLD/GANDALF) is small compared to
495 LP-DOAS/GANDALF. This is expected as the LP-DOAS is not an in situ technique and instead measures an average concentration along the light path. The ratios CRDS / GANDALF and LP-DOAS/GANDALF are close to unity, whereas in the case of CE-DOAS/GANDALF and CLD/GANDALF they deviate from unity by 0.15. All ratios distribution generally show a trend close to a normal distribution (Fig. 11 [A1, A2, A3, and A4]) but the skewness in LP-DOAS / GANDALF (A1 in Fig. 11) on both sides of the average value is relatively largest. In the lower panel of Fig. 11 [B1, B2, B3, and B4], the probabilities show
500 a deviation from normality and a tail on top (towards the right) and bottom (towards the left) sides can be observed. The tail could be an indicator of outliers, caused by for example the non-normality of the precision at low values, background level, and potential interferences of $NO_2$ instruments. The lower panels [B1, B2, B3 and B4] of Fig. 11 show that a major fraction of the ratios is normally distributed, evident from the $25^{th}$ to $75^{th}$ interquartile range of probability in all cases. The percentile of probability towards normality is slightly greater (about $10^{th}$ to $90^{th}$ percentile) in the case of CLD/GANDALF compared to the
505 others. The percentile is about $15^{th}$ to $80^{th}$ and $25^{th}$ to $90^{th}$ with (CRD/GANDALF, CE-DOAS/GANDALF) and (LP-DOAS/GANDALF), respectively. A perfect normal distribution should not be expected in these cases as mathematically a ratio between two normally distributed quantities does not follow a normal-distribution but it can be a distribution like the Cauchy-distribution (Weisstein, 2003). The long tails in the lower panel of Fig. 11 [B1, B2, B3, and B4] also indicates characteristics of the Cauchy distribution. In this type of distribution, the accuracy of average and standard deviation values cannot be increase by
510 increasing the number of data points.

 To identify systematic deviations based on other trace gases or parameters, ratios are further compared with the observed data of several trace gases, radiation, and meteorological parameters. There are only two cases, where a systematic correlation of ratios was observed with the observed quantities during PARADE, as shown in Fig. 12, and Fig. 13. In **Case 1**, ratios are presented as a function of the observed $O_3$ concentrations. The ratio between CLD and GANDALF shows a decreasing
515 trend with respect to an increase in the $O_3$ concentrations (subplot **C4**, Fig. 12). This ratio (CLD/GANDALF) averages 0.95 at levels less than 20 ppb $O_3$. It decreases to an average of 0.86 over the interval of 20 to 42 ppb $O_3$, while averaging 0.81 at levels above 42 ppb of $O_3$. There is no trend observed in other ratios (CRDS/GANDALF, LP-DOAS/GANDALF, and CE-DOAS/GANDALF) as shown in Fig. 12. The subplot (**C4**, Fig. 12) has been cross-checked by altering the GANDALF data in the denominator to the other three measurements (LP-DOAS, CRDS, and CE-DOAS) and qualitatively similar trends were
520 observed as with GANDALF. The reason for this CLD/GANDALF trend is not clear at the moment. However, it seems that this trend may be an indirect impact due to the zero-air measurement of the CLD with the BLC unit ON which is dependent on the

converter's history (exposition to ambient NO, NO$_2$, and HNO$_3$ concentrations along humidity) and potentially affect the ambient NO$_2$ measurements. So the dependency on O$_3$ might be an indirect effect: high ozone could point to transport from above with lower H$_2$O and lower NO$_x$, which both could affect the zero leading to an overestimation of the subtracted zero signal. In **Case 2** (subplot **D3**, Fig. 13), a correlation is observed for the ratio between CE-DOAS/GANDALF as a function of jNO$_2$. At higher values of jNO$_2$, the ratio approaches unity. The sampling line for CE-DOAS and CRDS was the same and no correlation for the ratio between CRDS and GANDALF is seen with respect to jNO$_2$. However, the data for the CRDS instrument was corrected for the effect of 'NO + O$_3$ → NO$_2$' in the sampling line and this correction for the CE-DOAS instrument was not implemented. Hence, the jNO$_2$ trend in the ratio could be indirectly from 'NO + O$_3$'. A residence time of 10 s in the sampling line for the 'NO + O$_3$ → NO$_2$' reaction (using measured NO, and O$_3$) is sufficient to explain this trend. This correlation is also not observed for the ratios of LP-DOAS and CLD with respect to GANDALF. A cross check was done for panel D3 (Fig. 13) by exchanging GANDALF in the denominator to three other measurements (LP-DOAS, CRDS, and CLD); qualitatively similar trends were observed as previously. Besides the above-described systematic correlations, no indication of a potential interference is obtained for any of the instruments.

**4 Summary**

The laser-induced fluorescence based instrument (GANDALF) has been developed for the measurement of atmospheric NO$_2$. GANDALF has been tailored towards compact design with a low detection limit (5 – 10 ppt 1min$^{-1}$), and high precision (0.5% + 3 ppt 1min$^{-1}$), making it capable of measuring NO$_2$ throughout the troposphere with a time resolution of 1 minute. The reliability of GANDALF was successfully tested during the PARADE-2011 field campaign. Several available NO$_2$ measurements based on different methods (absorption spectroscopy, chemiluminescence, and fluorescence) provided a unique chance of successful inter-comparison. In general, all instruments performed well. GANDALF showed a very good correlation ($R^2 \approx 0.99$) in comparison to other in situ instruments (Fig. S11 in the supplement), and even with LP-DOAS the correlation was $R^2 \approx 0.9$. The differences in the absolute values were within the specified range of individual measurement errors. The main advantages and disadvantages of GANDALF compared to the other instruments are summarized as follows.

In comparison to the CRDS instrument, the main advantage for GANDALF is that the sampling can be achieved without an inlet-line. This is not possible for the close-path CRDS system. This provides the capability of the detection at ambient temperature for GANDALF, which is especially of an advantage for aircraft measurements of NO$_2$ where avoiding interference from CH$_3$OONO$_2$ and HO$_2$NO$_2$ (via unwanted thermal dissociation) is very important. The requirement of calibration is the main disadvantage for GANDALF compared to CRDS (absolute technique). However, both instruments require frequent zero-air measurements. The limit of detection for both instruments was of similar magnitude during PARADE-2011.

The CE-DOAS instrument is comparable to the CRDS instrument. It also needs frequent background measurements but no absolute calibration. GANDALF has a much better in the sensitivity compared to the CE-DOAS instrument. During PARADE-2011, the detection limit for CE-DOAS was around 300 ppt (2σ, 30 s) while for GANDALF the detection limit was 5 – 10 ppt (min$^{-1}$). A low-cell-pressure is typically required to achieve a good sensitivity for LIF instruments (Table 1) while the detection in the other instruments (CRDS and CE-DOAS) is performed at sub-ambient pressures (>800 hPa). The requirement of calibration and usage of a larger scroll-pump (to achieve a low-cell-pressure) adds extra effort/cost to the GANDALF measurements.

The basic requirements for a calibration and background measurements are same in CLD and GANDALF. In the case of CLD, the maintenance is relatively easy compared to GANDALF. But GANDALF provides a direct detection of NO$_2$ compared

to the indirect detection of $NO_2$ (via $NO_2 \rightarrow NO$) in the CLD instrument. The sensitivity of GANDALF was better than the CLD instrument during PARADE-2011.

LP-DOAS does not require calibration or the zero-air measurement. For this reason, the uncertainty of the data is also very small compared to GANDALF or other in situ measurements. This is the main advantage of the LP-DOAS instrument over GANDALF. The restriction of this method is that it does not provide a local measurement. Also, the temporal resolution is
565 limited compared to other in situ instruments. The sensitivity of the LP-DOAS instrument generally depends on the length of a light path, and variations in visibility. It was on average about 110 ppt ($2\sigma$, 11 s) during PARADE-2011.

The selectivity of $NO_2$ measurement with GANDALF compared to other measurements in ambient air was assessed during PARADE and no potential interference was found. This prototype could provide useful measurements of $NO_2$ under remote conditions where an interference-free detection is absolutely essential for the study of $NO_x$ chemistry especially in the
570 context of $O_3$ formation, and radical loss processes.

**Outlook:** $NO_2$ in the free troposphere is variable (seasonally) and generally lower than 50 ppt (Gil-Ojeda et al., 2015). Depending on the location, in the free troposphere and the marine boundary layer, $NO_2$ can be as low as a few ppt (Beygi et al., 2011;Schreier et al., 2016). These $NO_2$ ranges are below the detection limit for the instrument (GANDALF) for short time resolutions of 1s, for example. Improvements for future use on aircraft are possible by further reducing the background of the
575 instrument. Since most of the background signal is from the fluorescence contamination of the Herriot's cell mirrors, this could be avoided by using a single beam (as demonstrated by (Di Carlo et al., 2013)) of the laser for detection without a Herriott cell or by using different coatings on the Herriott cell mirrors to increase reflectivity and reduce fluorescence. The current CW diode laser of the instrument may be replaced by an already available mono-mode dual diode laser [$\lambda$ (online) = 445 nm and $\lambda$ (offline) = 442 nm] for on and off resonance measurements of $NO_2$. Replacement of the current laser by a dual diode laser will decrease
partially the dependency on the frequent zero-air background measurements.

The formation of $RONO_2$ is an important sink for $NO_x$ and effects the ozone production efficiency (Browne and Cohen, 2012). The accurate measurement of $RONO_2$ is important for the assessment of local $O_3$ abundances. LIF systems in combination with the thermal dissociation method (Day et al., 2002) are also used and very useful for the detection of $RONO_2$, $RONO_2$, and $HNO_3$. GANDALF will be capable (currently under development) of measuring these species by coupling with the
585 thermal dissociation inlets. This further development could provide very useful data in the future to constrain models.

## 5 Acknowledgements

This work was done as a part of the first author's PhD, who is grateful for the constructive comments of Prof. P. Hoor during the
590 PhD advisory committee meetings. The financial support from DFG (Deutsche Forschungsgemeinschaft) within the ''DFG-Verfahren: Schwerpunktprogramm, SPP 1294: Bereich Infrastruktur - Atmospheric and Earth system research with the "High Altitude and Long Range Research Aircraft" (HALO)'' is gratefully acknowledged. The authors are thankful to M. Tang, B. Bohn, F. Berkes, and G. Phillips, for the data of $NO_3/N_2O_5$, $jNO_2$, $H_2O$, and $ClNO_2$, respectively. The acknowledgement extends to K. Hens, A. Novelli, E. Regelin, C.T. Ernest, C. Mallik for the useful comments/logistics, the site engineers, DWD
(Germany's National Meteorological Service) for meteorological data, and to the Goethe University, Frankfurt, for use of the Taunus Observatory facilities. We are thankful to the three anonymous referees for their comments and suggestions that helped us to improve the draft. We are also thankful to Lisa Whalley (editor) for the review process.

**Data availability:** Details about the field campaign can be found at http://parade2011.mpich.de/. The data related to PARADE-2011 can be obtained on request (by Hartwig Harder) from the responsible persons/owners.

**Competing interests:** There is no conflict of interests to declare.

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

**7 Tables**

**Table 1: Overview of different LIF instruments**

| Reference | $\lambda^{\text{laser type}}$ (nm) | Laser power (mW) | Absorption cross-section ($\times 10^{-19}$) cm$^2$ molecule$^{-1}$ | Cell pressure (Pa) | LOD (ppt min$^{-1}$) |
|---|---|---|---|---|---|
| (George and Obrien, 1991) | 532 [1] | 250 | 1.5 | 37 | 600 |
| (Fong and Brune, 1997) | 565 [2] | 250 | 0.6 | 1000 | 460 |
| (Thornton et al., 2000) | 585 [3] | 100-400 | 1 | 467 | 6 |
| (Matsumi et al., 2001) | 440 [4] | 100 | 7 | 35 | 12 |
| (Matsumoto et al., 2001) | 523.5 [5] | 360 | 1.4 | 93 | 125 |
| (Cleary et al., 2002) | 640.2 [6] | 16 | 3.9 [C] | 27 | 145 |
| (Matsumoto and Kajii, 2003) | 532 [7] | 6500 | 1.5 | 267 | 4 |
| (Taketani et al., 2007) | 410 [8], 473 [9] | 10, 15 | 6, 3 | 67 | 390, 140 |
| (Parra and George, 2009) | 406.3 [10] | 35 | 6 | Ambient | 2000 [A] |
| (Dari-Salisburgo et al., 2009) | 532 [11] | 8000-12000 | 1.5 | 60 | 12 |
| (Di Carlo et al., 2013) | 532 [12] | 38000 | 1.5 | 533 | 9.8 (s$^{-1}$) |
| GANDALF | 447- 450 [13] | Max. 200 | 5.3 [E] | 700 | 5-10 |

[E] Effective absorption cross-section; [C] Cooling enhancement; [A] Ambient pressure in the detection cell.

**Laser type (**Table 1 column 2**)**

[1] Nd: YAG laser; [2] Copper vapour laser-pumped dye laser; [3] Pulsed YAG-pumped dye laser; [4] Optical parametric oscillator laser; [5] Nd: YLF laser harmonic; [6] External-cavity tunable diode laser; [7] Nd:YVO$_4$ pulse laser pumped by a solid-state laser; [8] GaN-based laser diode; [9] Diode-pumped Nd:YAG laser; [10] CW GaN semiconductor laser diode; [11] YAG Q-switched intra-cavity doubled laser; [12] YAG Laser (Nd:YVO4 pulse laser); [13] CW diode laser

**Table 2: NO$_2$ instruments located or sampling at the top of the platform during PARADE-2011.**

| Measurement (Operator) | Technique | Uncertainty | Detection Limit | Time resolution |
|---|---|---|---|---|
| **LP-DOAS** (IUP-HD) | Long Path DOAS | 2 % | Avg.= 110 ppt (11s, 2σ) | 13 s |
| **CE-DOAS** (IUP-HD) | Cavity-Enhanced DOAS | 5 - 10 % | 300 ppt (30 s, 2σ) | 2 s |
| **CRDS** (MPIC) | Cavity Ring-Down Spectrometer | 6 % ; 20 ppt | 50 ppt (4s, 2σ) | 4 s |
| **CLD (BLC)** (MPIC) | Chemiluminescence Detector/ Blue light convertor | 105 ppt; 10 % | 55 ppt (2 s, 1σ) | 2 s |
| **GANDALF** (MPIC) | Laser-Induced Fluorescence | 5 % + 11 ppt (1σ) | 5 - 10 ppt (1 min, SNR = 2) | 1 s |

**Table 3: Fit parameters based on the bivariate model function according to the relation ($NO_2$Instruments = a × [Median$NO_2$] + b) at different $NO_2$ ranges. The value of $NO_2$Instruments-intercept 'b' is in ppb. 'N' is number of data points and $R^2$ is the square correlation coefficient. ± δ is the standard error of slope 'a' and intercept 'b'.**

| $NO_2$ Instruments | a | ± $\delta_a$ | b | ± $\delta_b$ | N | $R^2$ | a | ± $\delta_a$ | b | ± $\delta_b$ | N | $R^2$ |
|---|---|---|---|---|---|---|---|---|---|---|---|---|
| | $NO_2 < 1$ppb | | | | | | $NO_2 \geq 1$ to $\leq 6$ ppb | | | | | |
| LP-DOAS | 1.23 | 0.07 | -0.15 | 0.05 | 208 | 0.80 | 1.03 | 0.008 | -0.03 | 0.01 | 964 | 0.90 |
| CE-DOAS | 0.95 | 0.06 | -0.06 | 0.05 | 208 | 0.80 | 0.92 | 0.01 | -0.03 | 0.02 | 964 | 0.99 |
| CRDS | 1.1 | 0.07 | -0.02 | 0.05 | 208 | 0.83 | 1.06 | 0.01 | 0.002 | 0.02 | 964 | 0.99 |
| CLD | 0.99 | 0.08 | -0.12 | 0.06 | 208 | 0.73 | 0.97 | 0.01 | -0.13 | 0.02 | 964 | 0.98 |
| GANDALF | 1.06 | 0.07 | 0.015 | 0.05 | 208 | 0.74 | 1.04 | 0.01 | 0.015 | 0.02 | 964 | 0.99 |
| | $NO_2 > 6$ to $< 12$ ppb | | | | | | $NO_2 \geq 12$ ppb | | | | | |
| LP-DOAS | 1.2 | 0.08 | -1.51 | 0.6 | 52 | 0.64 | 1.42 | 0.2 | -6.64 | 4 | 15 | 0.69 |
| CE-DOAS | 0.91 | 0.09 | 0.075 | 0.7 | 52 | 0.94 | 0.87 | 0.2 | 0.55 | 3 | 15 | 0.96 |
| CRDS | 1.09 | 0.09 | -0.16 | 0.6 | 52 | 0.94 | 1.04 | 0.2 | 0.38 | 3 | 15 | 0.94 |
| CLD | 1.02 | 0.1 | -0.64 | 0.7 | 52 | 0.81 | 0.89 | 0.2 | 0.51 | 3 | 15 | 0.84 |
| GANDALF | 1.05 | 0.08 | 0.016 | 0.6 | 52 | 0.94 | 0.99 | 0.2 | 0.52 | 3 | 15 | 0.94 |

**Table 4: Average values of $NO_2$ ratios during PARADE-2011. These are derived from the different overall-$NO_2$-measurements with respect to GANDALF overall-$NO_2$-measurement.**

| Ratio | Average | Standard Deviation |
|---|---|---|
| LP-DOAS / GANDALF | 0.96 | 0.19 |
| CRDS / GANDALF | 1.01 | 0.06 |
| CE-DOAS / GANDALF | 0.86 | 0.07 |
| CLD / GANDALF | 0.85 | 0.09 |

**8 Figures**

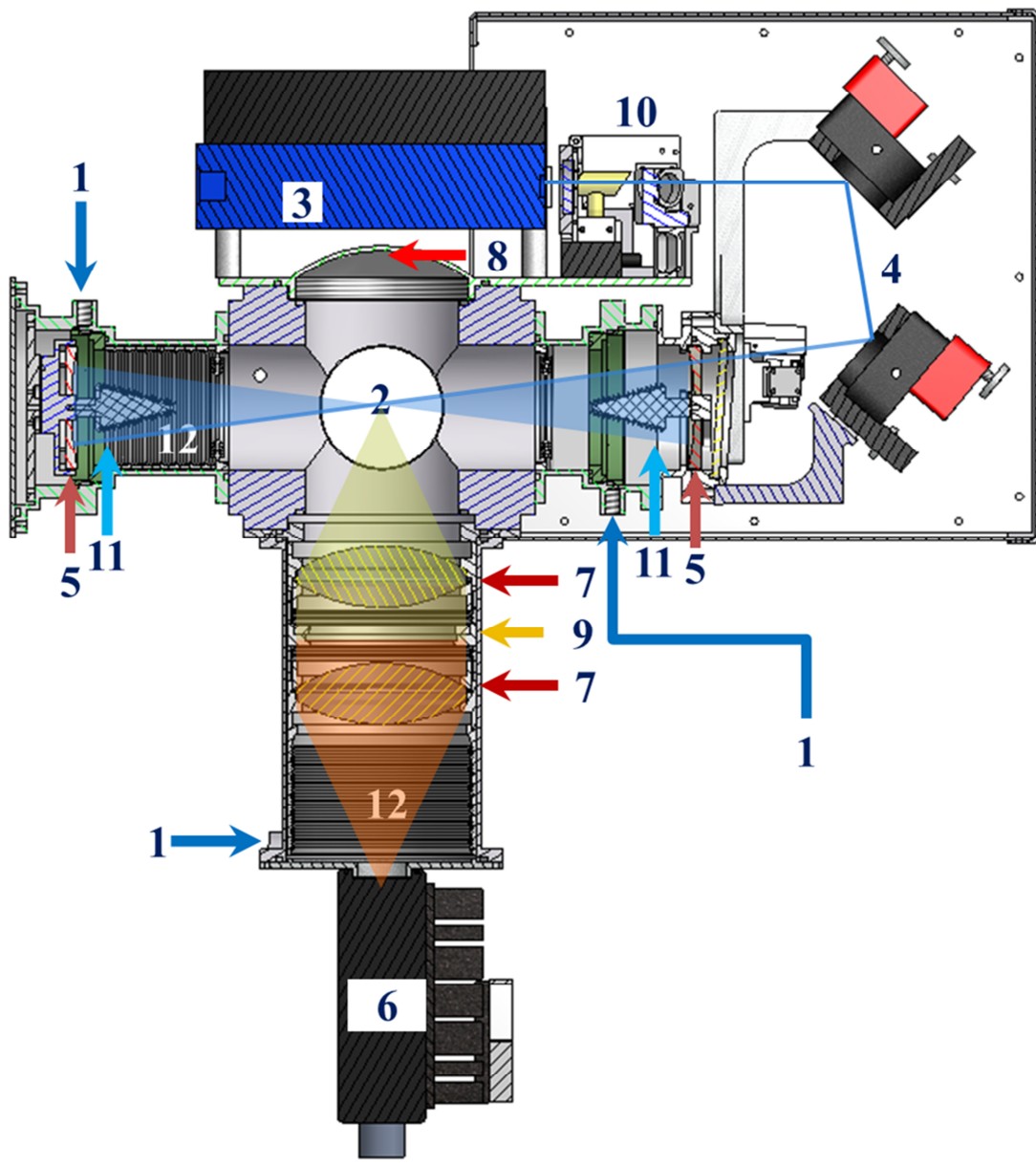

830

**Figure 1: Section view[21] of GANDALF**

**(1):** Flushing for optics **(2):** detection area **(3):** Diode laser **(4):** Motorised mirrors **(5):** Herriott cell's mirrors **(6):** PMT **(7):** Focusing lens **(8):** Concave mirror **(9):** Interference/optical filters **(10):** Optical reference system **(11 and 12):** Baffles

835

---

[21] Section view is based on Inventor-2009: The figure is created by defining a plane used to cut through the whole assembly. 3D AutoCAD models (1) for the diode laser by courtesy of Omicron Laserage Laserprodukte GmbH and (2) for optical mirror holders by courtesy of Newport.

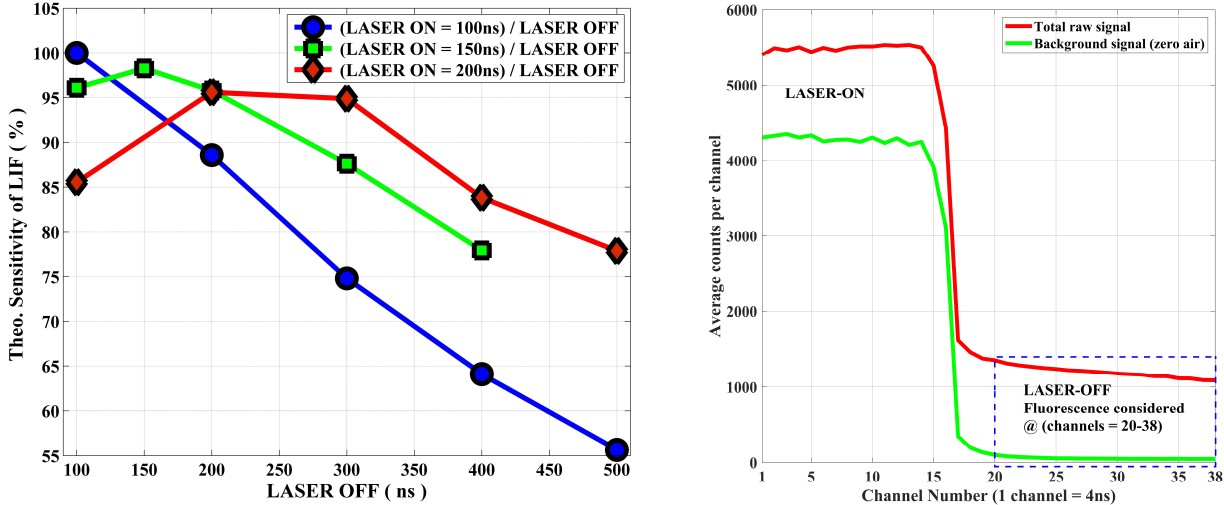

**Figure 2: (Left-side): Relative sensitivity of the instrument based on simulation is demonstrated for three different on/off cycles of diode laser operation. (Right-side): ON-OFF cycle of the laser for a signal of about 12ppb NO$_2$ [y-axis arb. unit] is shown. The sum of channels (20-38) is considered as fluorescence signal for the data analysis. A schematic of the data acquisition system is provided in the supplement.**

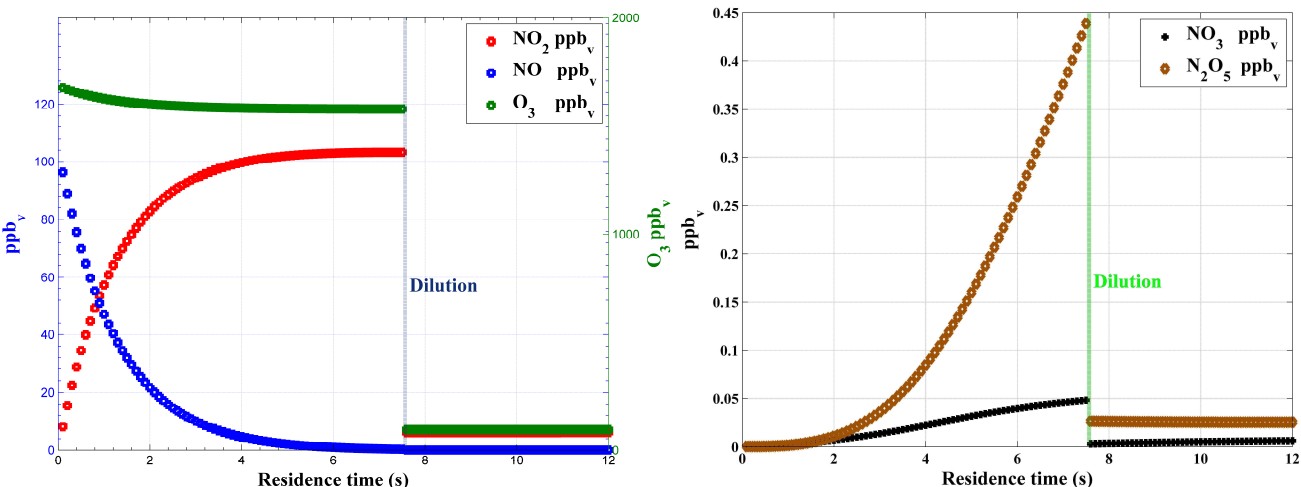

**Figure 3: Box model simulation of gas phase titration of NO and O$_3$ (left panel) with loss of NO$_2$ due to formation of NO$_3$ and N$_2$O$_5$ (right panel).**

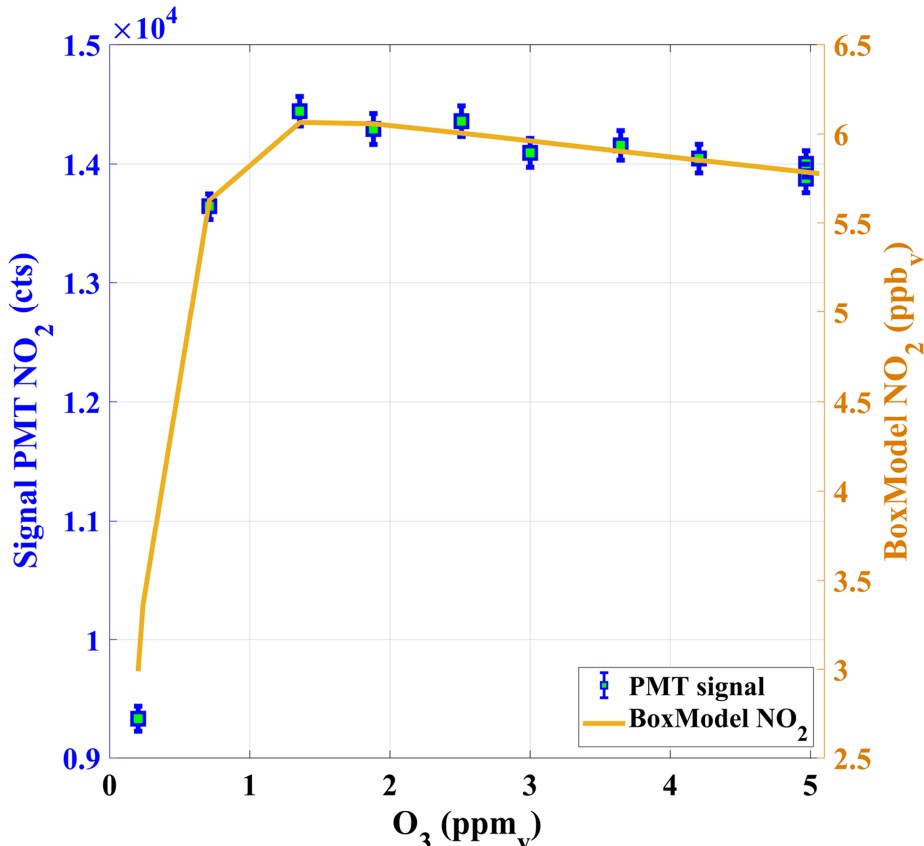

Figure 4: The PMT NO₂ signals in counts (cts) are shown as a function of O₃ concentrations in the calibrator (y-axis scale on the left side), together with NO₂ calculated from a box model of the NO₂ production in the calibrator (y-axis scale on the right side).

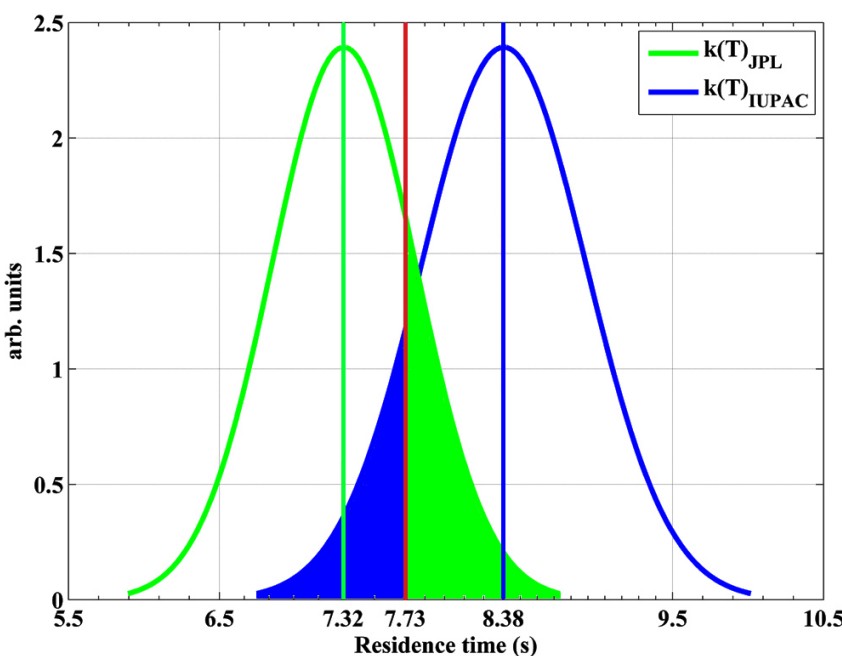

Figure 5: Residence time for NO₂ calibration gas in the calibrator based on Eq. 3. Also theoretically calculated residence time (7.73 s) is shown (red-line). The likelihoods (green or blue shaded areas) of residence times based on the JPL or IUPAC rate constant for being accurate are indistinguishable in comparison to the theoretically calculated residence time.

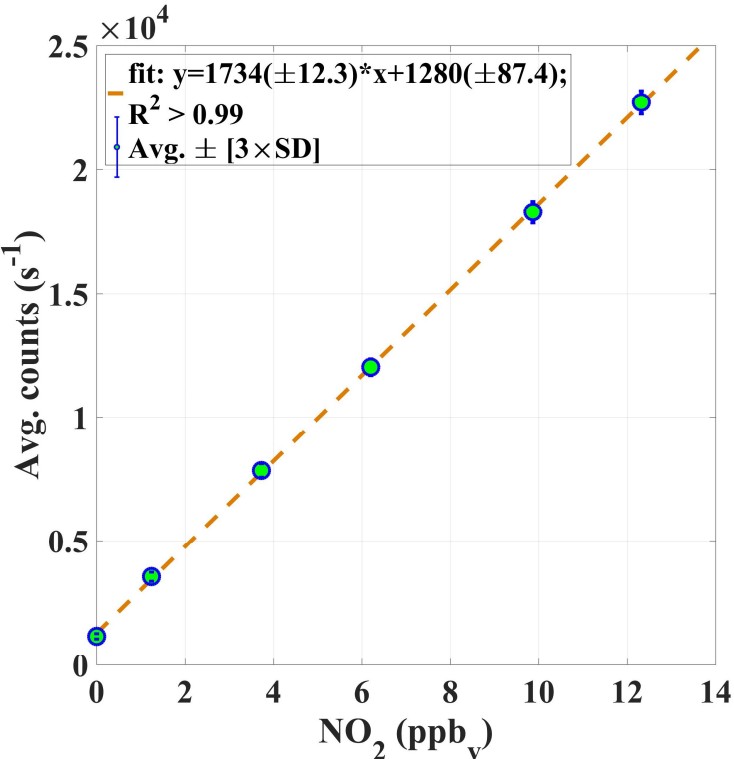

 **Figure 6: An example of the average PMT signal in counts (s⁻¹) vs known mixing ratios of NO₂. The calibration constant αc (Eq. 1) is given by the slope of the curve.**

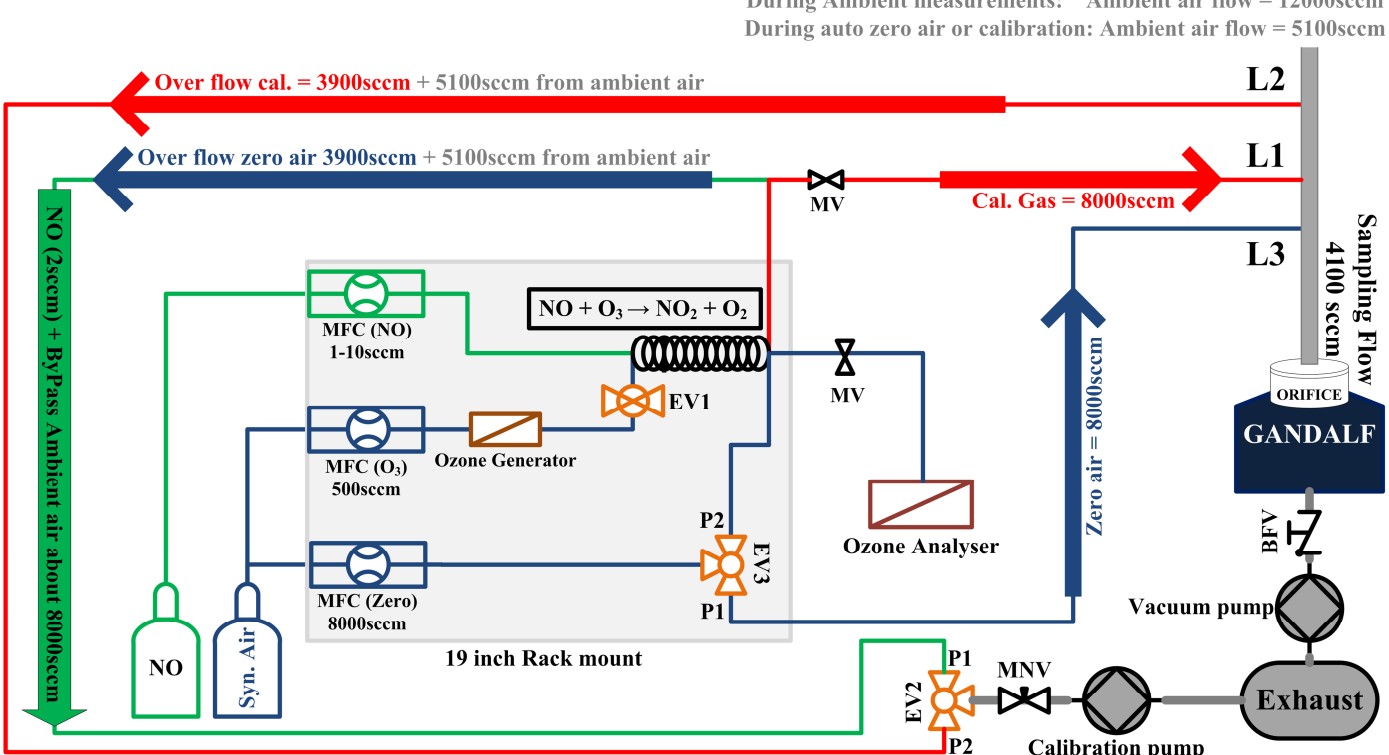

 **Figure 7: Schematic setup for the automated calibrations during PARADE-2011.**

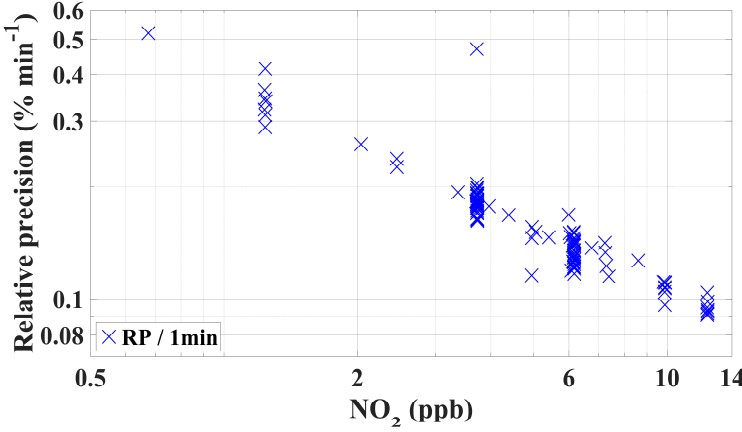

**Figure 8: The relative 1 minute precision of GANDALF is shown for PARADE-2011 as a function of NO₂ mixing ratios. The relative precision is calculated from randomly selected PMT signals during different calibration periods.**

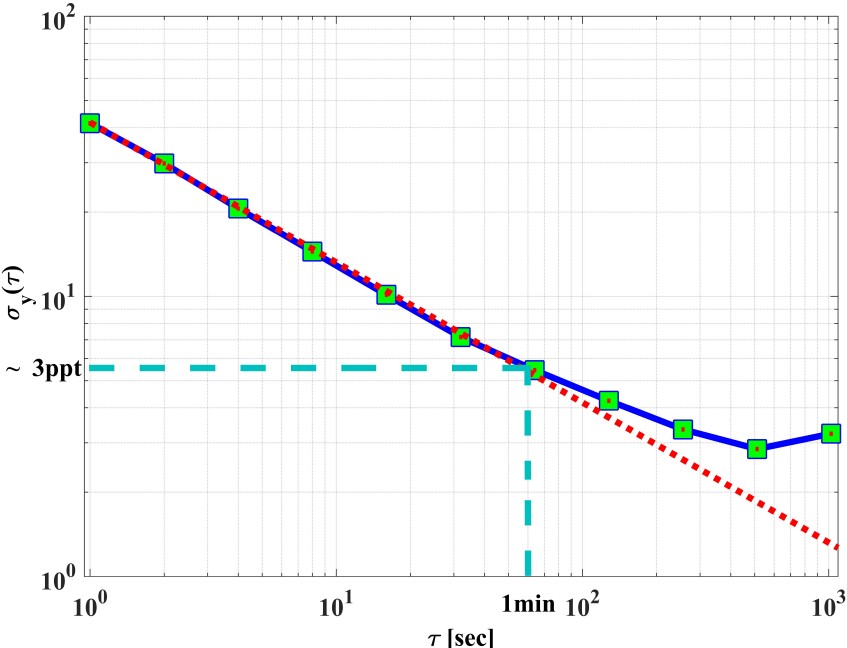

**Figure 9: An overlapping Allan deviation plot for the dependence of the 1σ variation in the background signal vs. integration time. The red colour dotted line shows the square root dependency of the signal.**

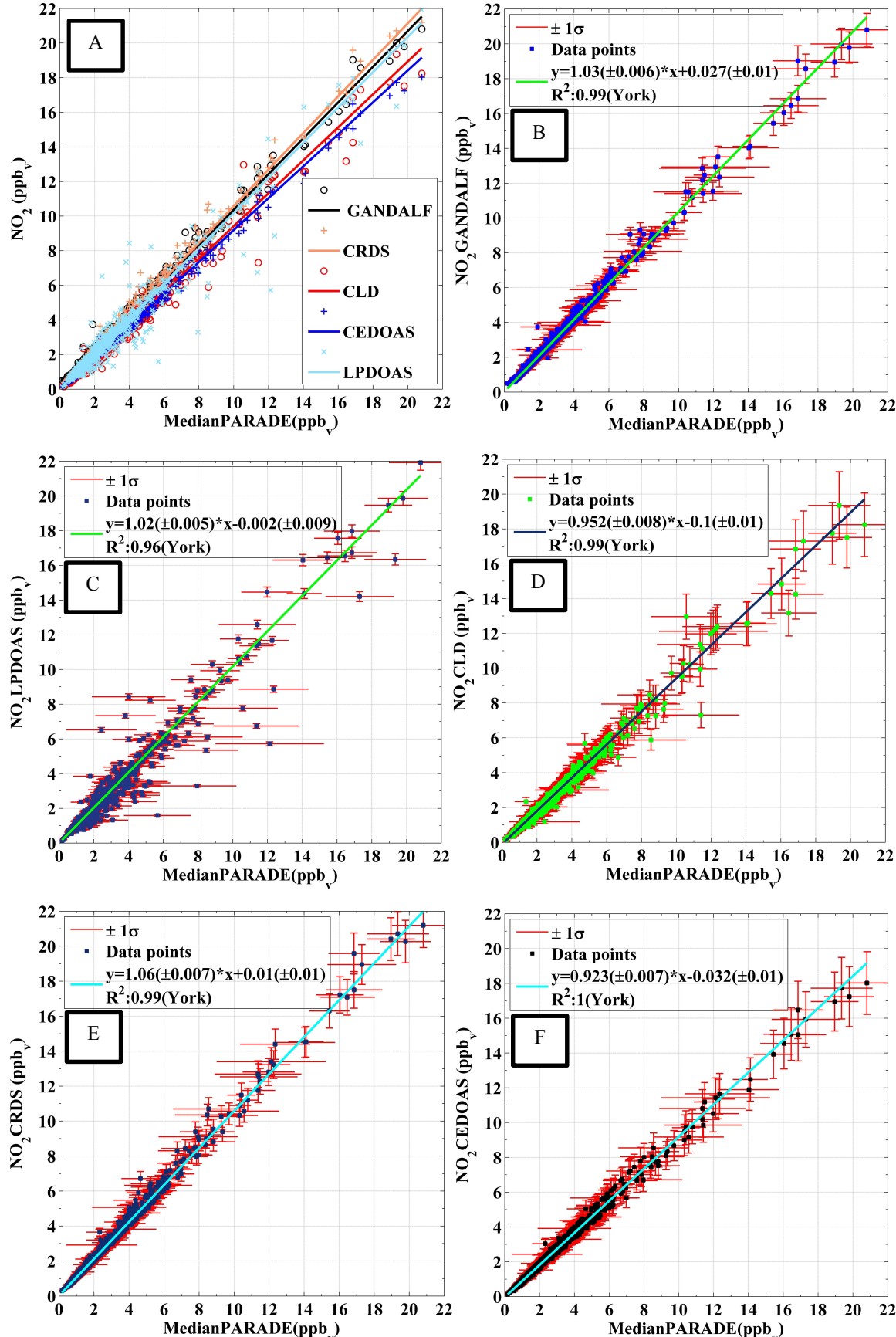

**Figure 10: Correlation plots of individual NO₂ measurement versus the derived median values of all NO₂ measurement at the platform during PARADE. [A]: Overall, [B]: GANDALF, [C]: CLD, [D]: CRDS, [E]: LP-DOAS, [F]: CE-DOAS**

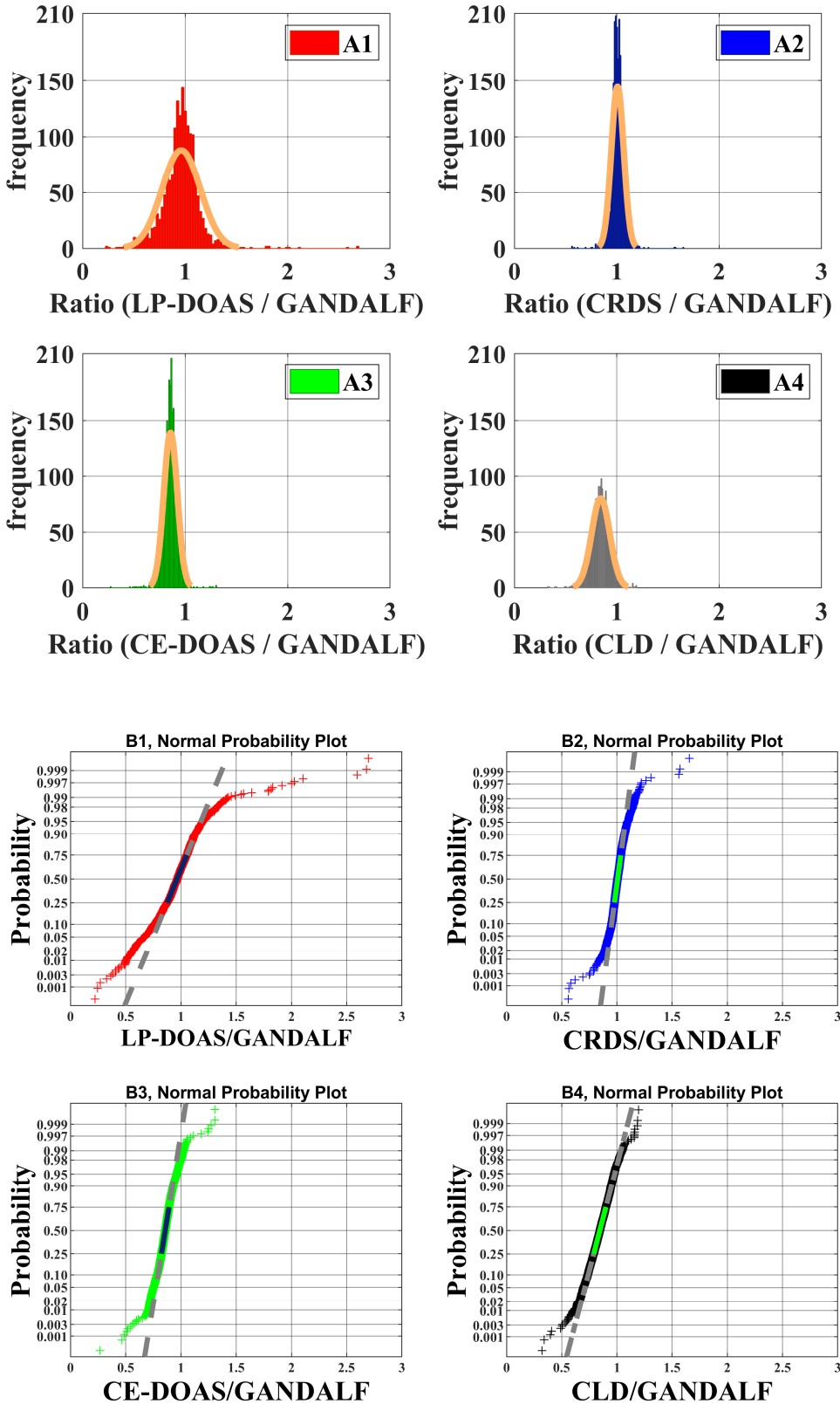

**Figure 11: Distribution of comparative instrument ratios of NO₂ measurements from different instruments is shown in upper panels (A1→A4) and a normal probability plot for comparative instrument ratios is shown in lower panels (B1→B4).**

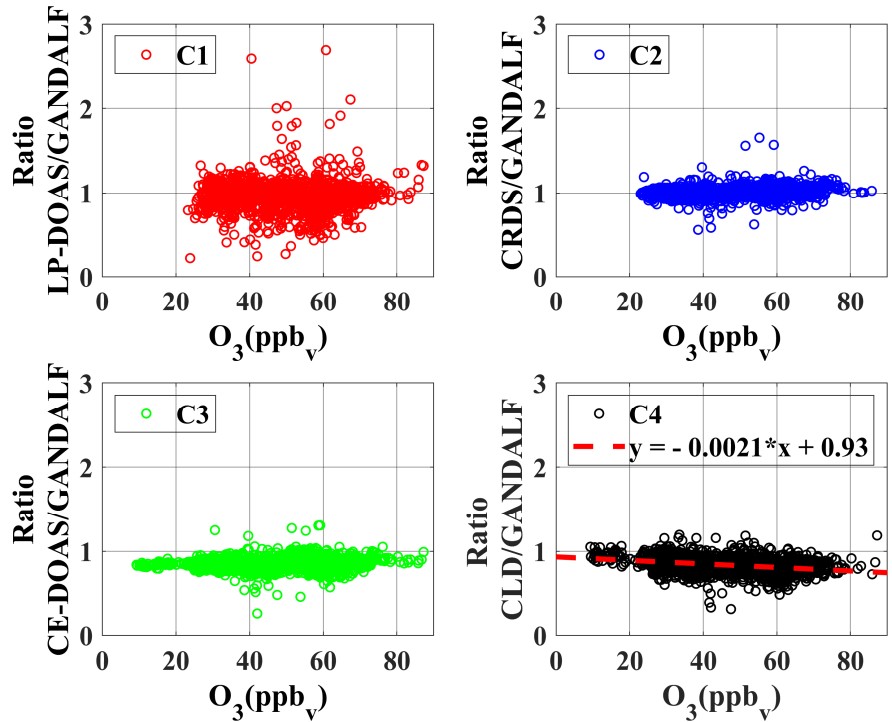

Figure 12: Ratios as a function of ambient $O_3$ during PARADE.

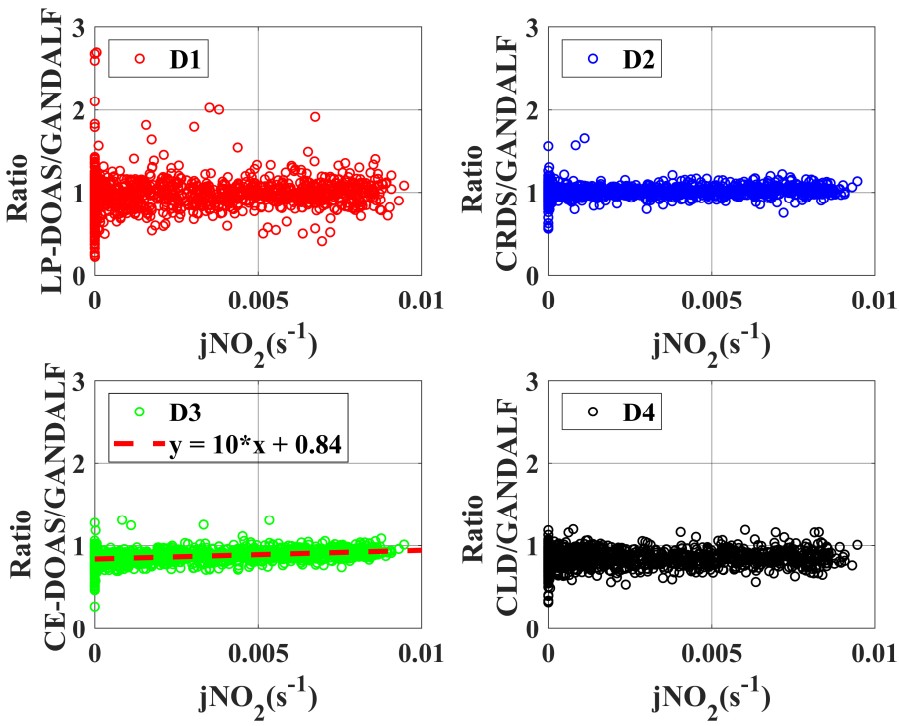

Figure 13: Ratios as a function of measured $jNO_2$ during PARADE.