# Peer review of "Laser induced fluorescence based detection of atmospheric nitrogen dioxide and comparison of different techniques during the PARADE 2011 field campaign"

_Atmospheric Measurement Techniques, 2018_

## Referee Comment (RC1) · Anonymous Referee #1 · 2 Aug 2018

General comments:
This manuscript describes a new laser induced fluorescence instrument developed for ground-based and aircraft measurements of NO2. The authors report the instruments characteristics, laboratory tests, an extensive description of the calibration system, and the first results of a field campaign in 2011, where they carried out an intercomparison with other systems that measured NO2 using different techniques.

The manuscript is generally well written, and the main results of the intercomparison

and the description of a new LIF system characteristics, which uses a CW laser is of a certain importance for future developer of NO2 systems and for the all community working on NOx measurements. In my opinion it fits with AMT scopes and I recommend publication, after the authors address the following questions and comments.

Specific comments:

lines 52-67: Since the aim of the manuscript is to describe a new NO2 instruments, and because there are different techniques to measure NO and NO2, I would limit the review of the measurements techniques to those for NO2 observation, omitting those for NO detection.

Line 150-155: I suggest to describe with more details the time-resolved florescence signal detection, trigger system, synchronization, how to take care of laser power fluctuation and so on, since this is the key part of the system that may be managed carefully using a CW laser.

Line 157-265: The calibration system that uses the NO tritation by O3 to produce NO2 is described and used in different ground-based instruments (i.e. Ryarson et al, 2000, Matsumoto et al., 2000, Osthoff et al., 2006). In my opinion it is a good approach that can be a system for periodical laboratory check of the instrument performance and of the possible NO2 cylinder degradation, but according also to figure 7 it is the bigger part of the system and includes many components not so compact such us the ozone generator and the ozone analyser. The use of this calibration system seems not easy on ground-based field campaign and really complicated on aircraft.

Technical corrections:

Line 52: It is quite rare but sometimes NO2 can be more than 100 ppb so I would replace '100' with 'hundreds'. Line 73: Add 'the' between 'in' and 'past'.

Line 74: remove the subscript to the 'v' of pptv.

Line 79: The reference reported (Dari-Salisburgo et al, 2009) describes the first

ground-based system developed by that group, I suggest to substitute this reference with the work of the same group (Di Carlo et al., 2013) that reports the evolution of their TD-LIF for aircraft measurements that has better sensitivity and performances.
Line 182: Remove 'Figure 3'
Line 710 (Table 1): I would include the evolution of the instrument described by Dari-Salisburgo et al, 2009, used also for aircraft measurements, because it uses another laser a Nd:YVO4 pulse laser, and has better performance in terms of LOD compared with that described in Dari-Salisburgo et al, 2009, more details can be found in Di Carlo et al., 2013.

Reference
Di Carlo P., E. Aruffo, M. Busilacchio, F. Giammaria, C. Dari-Salisburgo, F. Biancofiore, G. Visconti, J. Lee, S. Moller, C. E. Reeves, S. Bauguitte, G. Forster, R. L. Jones, and B. Ouyang, Aircraft based four-channel thermal dissociation laser induced fluorescence instrument for sim-ultaneous measurements of NO2, total peroxy nitrate, total alkyl nitrate, and HNO3, Atmos. Meas. Tech., 6, 971–980, 2013.
Osthoff, H. D., Brown, S. S., Ryerson, T. B., Fortin, T. J., Lerner, B. M., Williams, E. J., Pettersson, A., Baynard, T., Dube, W. P., Ciciora, S. J., and Ravishankara, A. R.: Measurement of atmospheric NO2 by pulsed cavity ring-down spectroscopy, J Geophys Res-Atmos, 111, Artn D12305 Doi 10.1029/2005jd006942, 2006.
Matsumoto, J., et al., Direct measurement of NO2 in the marine atmosphere by laser-induced fluorescence technique, Atmos. Environ., 35, 2803–2814, 2001.
Ryerson, T. B., E. J. Williams, and F. C. Fehsenfeld, An efficient photolysis system for fast-response NO2 measurements, J. Geophys. Res., 105, 26,447– 26,461, 2000.

---

## Referee Comment (RC2) · Anonymous Referee #2 · 13 Aug 2018

The manuscript of Umar Javed and colleagues is an interesting study on a new laser induced fluorescence instrument for the measurement of atmospheric nitrogen dioxide. It describes the set-up of this instrument with an emphasis on its calibration system and the analysis of possible cross-sensitivities. An important part of the manuscript focuses on the results of an intercomparison field campaign where different nitrogen dioxide measurement techniques have been compared. Nitrogen dioxide is an important atmospheric trace gas and imposes quite some efforts to perform good measurements with different techniques. This manuscript makes a valuable contribution to improve ni-

trogen dioxide measurement techniques. I recommend this manuscript for publication; however there are some points that should be addressed before.

Specific comments:

L16: Abstract: I suggest that in the abstract the field campaign PARADE should be mentioned; also the location and time of the field campaign.

L 32: Aircraft emissions as well are directly affecting the free troposphere.

L 96: The wavelength of the laser is given but not the wave length of the fluorescence.

L 103: In this context: What is the definition of zero air?

L 125 / L 144: What is the PMT temperature? Is the PMT actively cooled? What causes the background signal? Would the background decrease, if the PMT was cooled down at temperatures below 0°C by an active cooling unit?

L 223: "... at a lower temperature..." Which temperature?

L 237: Is there an explanation for the change in sensitivity? What is the range of sensitivity change?

L 268: Figure 8 shows the relative precision obtained during different calibrations. But how exactly do you determine the relative precision? Does it include for example only the variability of the sensitivity or the variability of the background, etc.? Please describe in more detail.

L 318: R6 is not a valid chemical reaction (both sides of the reaction arrow should be balanced).

L 337: I agree the short residence time of the sample air inside the instruments minimizes the thermal decomposition of the respective species. But please give at least a few calculated lifetimes against thermal decomposition for the most important interfering gases that illustrate this statement.

L505: I suggest that in the summary the authors underline the main advantage and disadvantage of this measuring system, also in comparison with other measurement techniques. What is the future of Gandalf (besides LOTR)? Are there specific plans to use this instrument during other field campaigns?

L 508: The authors are mentioning that the instrument is capable for measurements throughout the troposphere with a time resolution of 1 s to 1 min. However, the whole preceding discussion has been focused on ground based measurements at a time resolution of 1 min. Also the concentration of NO2 in the free troposphere is much lower than in the boundary layer. LOD would increase significantly if you reduced the sampling time from 60 s to 1s. Please outline in short what improvements would be necessary to achieve this goal.

Technical Corrections:

L 298: . . . about 8 time higher than the cross section of ...

Tables:

Table 3: $\pm \delta$ – explanation in the caption is missing.

Table 4: Uncomplete caption - which ratios?

Figures:

In general the figure captions are often not sufficient in explaining the content of the figures.

Figure 1: The numbers in the caption of this figure have different orientations and do not facilitate the reading. All numbers should have the same conventional orientation (like the numbers "9, 10,. . ."?. "SF" - This is not quite consistently. All other objects of this figure stand for units of the instrument. "Sampling flow" is the gas stream into the instrument (I assume) and not part of the instrument. So it would be more suitable to write: Inlet orifice or sampling flow line, or. . ..

[Figure]

Figure 4: . . . as a function of O3 concentration in . . . Please explain "arb" in the y-axis label. The caption is incomplete; "Box-Model NO2" is not mentioned.

Figure 5: "also theoretically calculated residence time (7.73s). . ." I assume the red line in this figure is meant.

Figure 6: What do you mean by calibration signal? I assume it is the number of counts at the PMT?

L 250 – L 265 / Figure 7: The description in the text and in the figure caption is a little bit confusing and should be clarified. E.g. an ozone analyzer is shown in the figure but not mentioned in the text. In the text blue, red and white arrows are mentioned; in the figure you find additionally orange arrows. In the text only red arrows in L2 are mentioned, but there also white arrows found in L2. I assume that the valves EV3 and EV2 have to point at the position P1(P2) at the same time? Above the Gandalf-box in Figure 7 there are three times written "4100 sscm" in different colors and different orientation. As long as you do not discuss it explicitly in the text,one "4100 ssm" label is enough. Figure caption: "outdoor – operations"? - Better during field campaigns or during the PARADE field campaign.

Figure 8: JD = Julian Days. The formulation of this caption is a little bit unclear. Please improve.

Figure 12 and 13. "Ratios. . .." – The readability would be improved if you would write in the caption which ratio is meant. Please choose the same y-scale for all figures.

---

## Referee Comment (RC3) · Anonymous Referee #3 · 15 Aug 2018

This manuscript describes a laser induced fluorescence instrument that has been developed for ground and aircraft based measurements of NO2. It describes the setup of the instrument, its calibration and examines possible interferences. There is also a description of data taken with the instrument during a field campaign that involved a range of different NO2 detection techniques and a comparison of the different datasets is made. NO2 is a key atmospheric constituent and it is important to develop new direct measurement techniques for it, making this type of work very topical. The manuscript is generally well written and provides an important reference for others wishing to de-

velop an LIF instrument for NO2 detection. I recommend publication subject to the authors dealing with the following relatively minor comments.

Specific comments: On line 254 it is stated that frequent zero-air measurements are necessary to monitor changes in the background signal. This is important as presumably the addition of air with zero NO2 in it is the only way that the background signal of the instrument can be measured? I therefore think that more discussion should be had into this. Firstly, how is the zero air generated, is it just from a cylinder or is there some further scrubbing carried out? How do the authors know how much NO2 is in their zero air, has it been measured? Some discussion should be had as to how the quality of the zero air effects the accuracy of the instrument.

In the instrument description section I feel that a diagram showing the timing of the laser pulse, PMT detection and fluorescence signal would be beneficial. All the information is there in the paragraph but a diagram would makes things much clearer.

On line 144 it is stated that the PMT and laser are kept at a constant temperature by a water chiller but at what temperature? Would lowering the temperature help with reducing background. Or conversely if no cooling was present, which may make the instrument easier to operate on an aircraft, how would this affect instrument performance?

On line 199 it is stated that in the calibration system, about 99% of the NO is consumed by titration with O3 to produce the NO2. Why do the authors choose to titrate this much? Surely there is a danger that they could have more O3 than NO in the system and hence have the potential for secondary chemistry to affect the amount of NO2 present? Would it not be better to titrate around 80% of the NO to NO2?

In the abstract it is stated that the instrument could be used for airborne measurements of NO2 however there is very little discussion of this in the manuscript. There should at least be some discussion as to how the instrument precision and detection limit would change for 1 second averaging (which would be required for aircraft work) and how this

compares to the potentially lower levels of NO2 present in the free troposphere.

Technical: Figure 1: It would be clearer if the numbers were all the same way up on the page. Reaction 2: what is the wavelength of the fluorescence?

---

## Author Comment (AC1) · 17 Dec 2018

"**Anonymous Referee #1**"
"**General comments:**"
"This manuscript describes a new laser induced fluorescence instrument developed for ground-based and aircraft measurements of NO2. The authors report the instruments characteristics, laboratory tests, an extensive description of the calibration system, 10 and the first results of a field campaign in 2011, where they carried out an intercomparison with other systems that measured NO2 using different techniques. The manuscript is generally well written, and the main results of the intercomparison and the description of a new LIF system characteristics, which uses a CW laser is of a certain importance for future developer of NO2 systems and for the all community working on NOx measurements. In my opinion it fits with AMT scopes and I recommend publication, after the authors address the following questions and comments."

**Response:** We are thankful for the Anonymous Referee (#1) for the review and useful comments on the draft.

"**Specific comments:**"
"**Lines 52-67:** Since the aim of the manuscript is to describe a new NO2 instruments, and because there are different techniques 20 to measure NO and NO2, I would limit the review of the measurements techniques to those for NO2 observation, omitting those for NO detection."

**Response:**

We agree with the reviewer. Lines 55-67 (based on the discussion draft) are replace in the revised draft as follows

"The Photofragmentation Two-Photon Laser-Induced Fluorescence (PF-TP-LIF) (Sandholm et al., 1990;Bradshaw et al., 1999) 25 and chemiluminescence (Fontijn et al., 1970) methods are well known for direct in situ NO detection. In the past, an indirect detection of $NO_2$ with these techniques has been performed by converting $NO_2 \rightarrow NO$ via photolytic/catalytic process followed by NO detection. However, in the case of $NO_2$ to NO conversion, a potential interference from $NO_z$ species cannot be fully excluded for the $NO_2$ measurement, e.g. (Crawford et al., 1996;Villena et al., 2012;Reed et al., 2016)."

"**Line 150-155:** I suggest to describe with more details the time-resolved fluorescence signal detection, trigger system, synchronization, how to take care of laser power fluctuation and so on, since this is the key part of the system that may be managed carefully using a CW laser."

**Response:**

To explain the data acquisition following "sentences" are included in the draft.

"A counter card is used for the data acquisition. There is no need for synchronisation as the counter card itself triggers the laser pulse. The timing system is entirely controlled by an FPGA (field-programmable gate array), utilizing an external crystal oscillator of 20MHz nominal frequency with a stability of +/-2.5ppm over the temperature range of -30°C to +75°C. All internal frequencies are derived from this clock by means of a PLL (phase-locked loop) in the FPGA. The triggering occurs at a fixed rate of 5 Mhz. The delay caused by the length of the trigger cable (propagation delay of the pulse), the laser power supply unit, 40 propagation delays from detector to FPGA, etc. is compensated with a programmable delay for the data acquisition in the FPGA.

So the FPGA logic recognizes when it should start recording the data after it emitted the trigger pulse and waits the specified amount of programmed clock cycles after emitting the trigger."

The signal from the PMT is attained for both periods of the laser cycle (100 ns ON, 100 ns OFF). The card has more than 50 channels available for the PMT data. Each channel has a resolution of 4 ns. The integrated raw data is written in a bin
file for the sampling period. The sampling period is typically 1s. The $NO_2$ fluorescence signal is resolved in the post analysis of the raw data. To elaborate the raw signal, a figure for the time-resolved fluorescence is added as a subpart of 'Fig. 2'to the draft.

The power of the diode laser is monitored and recorded continuously/simultaneously by using a photodiode. Later, the $NO_2$ signal is generally normalized by using the photodiode signal. This is a regular approach for any LIF instrument and has been described previously by many studies. It is noteworthy that the impact of the correction was not significant during the field
campaign PARADE. This is because during PARADE-2011, frequent calibrations were performed. So any variability in the power of the laser was captured via the calibration.

"**Line 157-265:** The calibration system that uses the NO titration by O3 to produce NO2 is described and used in different ground-based instruments (i.e. Ryarson et al, 2000, Matsumoto et al., 2000, Osthoff et al., 2006). In my opinion it is a good
approach that can be a system for periodical laboratory check of the instrument performance and of the possible NO2 cylinder degradation, but according also to figure 7 it is the bigger part of the system and includes many components not so compact such us the ozone generator and the ozone analyser. The use of this calibration system seems not easy on ground-based field campaign and really complicated on aircraft."

**Response:**
The formation of $NO_2$ via the gas phase titration is very common approach used for the calibrations of $NO_x$ analysers. A reference to the previous study (Ryerson et al., 2000) is included. We agree with the reviewer that the bigger part of the instrument is the calibration system. However, most parts of the system (like MFCs, valves, reaction chamber, etc.) including the ozone generator is part of a single 19-inch rack mount (4RU). The ozone analyser is also 19-inch rack mount (4RU). So basically, two 19-inch rack mount and a small pump are required for the complete system.

The calibration system can be used to check degradation/changes in the concentration over a period of time in a $NO_2$ cylinder. We adapted such an approach in the past, but the day to day variation in different $NO_2$ cylinders was hard to track, since these cylinders showed unstable concentrations with low repeatability even within a short period of time. These checks were performed with different instruments (CLD, CRD, and GANDALF) during different periods of time.  In a short time scale (hours), the observed difference was within 3-13% for different $NO_2$ gas cylinders. Whereas for a longer period (months), the
differences were roughly up to 30%.  Therefore, to get a reliable signal, the gas phase titration is advantageous compared to the use of a $NO_2$ cylinder.

"**Technical corrections:**"

"**Line 52:** It is quite rare but sometimes NO2 can be more than 100 ppb so I would replace '100' with 'hundreds'. Line 73: Add 'the' between 'in' and 'past'."

**Response:**
Replacement is added.

"**Line 74:** remove the subscript to the 'v' of pptv."

**Response:**

It is done.

“**Line 79:** The reference reported (Dari-Salisburgo et al, 2009) describes the first ground-based system developed by that group,

85 I suggest to substitute this reference with the work of the same group (Di Carlo et al., 2013) that reports the evolution of their

TD-LIF for aircraft measurements that has better sensitivity and performances.”

**Response:**

Added the reference "(Di Carlo et al., 2013)" at this position.

**Response:**

It is done.

“**Line 710 (Table 1):** I would include the evolution of the instrument described by Dari-Salisburgo et al, 2009, used also for

95 aircraft measurements, because it uses another laser a Nd:YVO4 pulse laser, and has better performance in terms of LOD

compared with that described in Dari-Salisburgo et al, 2009, more details can be found in Di Carlo et al., 2013.”

**Response:**

The successor instrument is indeed better in sensitivity from the predecessor. The overview of the instrument described in (Di Carlo et al., 2013) is also added to Table 1 as suggested by the reviewer.

**References**

Bradshaw, J., Davis, D., Crawford, J., Chen, G., Shetter, R., Muller, M., Gregory, G., Sachse, G., Blake, D., Heikes, B., Singh, H., Mastromarino, J., and Sandholm, S.: Photofragmentation two-photon laser-induced fluorescence detection of $NO_2$ and NO:
Comparison of measurements with model results based on airborne observations during PEM-Tropics A, Geophys Res Lett, 26, 471-474, Doi 10.1029/1999gl900015, 1999.

Crawford, J., Davis, D., Chen, G., Bradshaw, J., Sandholm, S., Gregory, G., Sachse, G., Anderson, B., Collins, J., Blake, D., Singh, H., Heikes, B., Talbot, R., and Rodriguez, J.: Photostationary state analysis of the NO2-NO system based on airborne observations from the western and central North Pacific, J Geophys Res-Atmos, 101, 2053-2072, Doi 10.1029/95jd02201, 1996.

Di Carlo, P., Aruffo, E., Busilacchio, M., Giammaria, F., Dari-Salisburgo, C., Biancofiore, F., Visconti, G., Lee, J., Moller, S., Reeves, C. E., Bauguitte, S., Forster, G., Jones, R. L., and Ouyang, B.: Aircraft based four-channel thermal dissociation laser induced fluorescence instrument for simultaneous measurements of NO2, total peroxy nitrate, total alkyl nitrate, and HNO3, Atmos Meas Tech, 6, 971-980, 10.5194/amt-6-971-2013, 2013.

Fontijn, A., Sabadell, A. J., and Ronco, R. J.: Homogeneous Chemiluminescent Measurement of Nitric Oxide with Ozone -
Implications for Continuous Selective Monitoring of Gaseous Air Pollutants, Anal Chem, 42, 575-579, Doi 10.1021/Ac60288a034, 1970.

Reed, C., Evans, M. J., Di Carlo, P., Lee, J. D., and Carpenter, L. J.: Interferences in photolytic NO2 measurements: explanation for an apparent missing oxidant?, Atmos. Chem. Phys., 16, 4707-4724, 10.5194/acp-16-4707-2016, 2016.

Ryerson, T. B., Williams, E. J., and Fehsenfeld, F. C.: An efficient photolysis system for fast-response NO2 measurements, J
Geophys Res-Atmos, 105, 26447-26461, Doi 10.1029/2000jd900389, 2000.

Sandholm, S. T., Bradshaw, J. D., Dorris, K. S., Rodgers, M. O., and Davis, D. D.: An Airborne Compatible Photofragmentation 2-Photon Laser-Induced Fluorescence Instrument for Measuring Background Tropospheric Levels of No, Nox, and No2, J Geophys Res-Atmos, 95, 10155-10161, DOI 10.1029/JD095iD07p10155, 1990.

Villena, G., Bejan, I., Kurtenbach, R., Wiesen, P., and Kleffmann, J.: Interferences of commercial NO2 instruments in the urban
atmosphere and in a smog chamber, Atmospheric Measurement Techniques, 5, 149-159, DOI 10.5194/amt-5-149-2012, 2012.

---

## Author Comment (AC2)

**"Anonymous Referee #2"**

"The manuscript of Umar Javed and colleagues is an interesting study on a new laser induced fluorescence instrument for the measurement of atmospheric nitrogen dioxide. It describes the set-up of this instrument with an emphasis on its calibration system and the analysis of possible cross-sensitivities. An important part of the
manuscript focuses on the results of an intercomparison field campaign where different nitrogen dioxide measurement techniques have been compared. Nitrogen dioxide is an important atmospheric trace gas and imposes quite some efforts to perform good measurements with different techniques. This manuscript makes a valuable contribution to improve nitrogen dioxide measurement techniques. I recommend this manuscript for publication; however there are some points that should be addressed before."

*Response:* We appreciate the time given by the Anonymous Referee (#2) for the review. The helpful comments of the reviewer will provide more clarity to the draft.

**"Specific comments:"**
**"L16:** Abstract: I suggest that in the abstract the field campaign PARADE should be mentioned; also the location and
time of the field campaign."

**Response:**

The name of the campaign along location and time is added in the abstract.

**"L 32:** Aircraft emissions as well are directly affecting the free troposphere."

**Response:**

Aircraft emissions, as a source for the $NO_x$ in the upper troposphere (Strand and Hov, 1996), are included.

**"L 96:** The wavelength of the laser is given but not the wavelength of the fluorescence."

**Response:**

The $NO_2$ fluorescence has a broad spectrum. It starts at the excitation wavelength and extends into the infra-red region (Wehry, 1976). But still, the major fraction of the fluorescence still lies in the visible region (Sakurai and Broida, 1969;Sugimoto et al., 1982). In our case, we block the light with the interference filter at wavelengths < 550 nm. The cut-off band of the PMT is at about 890 nm. So the wavelength range of the detection window of the fluorescence signal is roughly between 550-890 nm. Still, we expect the major portion would be in the visible region. This is already explained in the text, so we have added also the lower
limit "$\lambda \geq 449nm$" for the emitted fluorescence in "R.2".

**"L 103:** In this context: What is the definition of zero air?"

**Response:**

The zero air is Synthetic air (hydrocarbon-free) as specified by Westfalen (99,999 mol% pure mixture of 20.5% $O_2$ in $N_2$, $H_2O$ <
5 ppmv, HC < 0.1 ppmv, $NO_x$ < 0.1 ppmv). Some studies e.g., (Thieser et al., 2016) showed that it is not fully free of $NO_2$

contamination, though levels are generally much smaller than the 0.1 ppmv specified. They estimate up to 20 ppt $NO_2$ in their supply of synthetic air cylinders. The Synthetic air was used during the PARADE field campaign for the background measurements of our instrument. The quality of the air is discussed in the section 3 (related to PARADE) of the draft.

Ideally, the zero air should be a replica of ambient-air but without the $NO_2$. In the past, it has been tried to use scrubbing techniques, based on active charcoal, coated surface with a certain chemical etc., to remove $NO_2$ from the ambient-air and use the scrubbed air for the background measurements (Matsumoto and Kajii, 2003). This can be done for a lower sampling rate, but at our high sampling flow rates (>4SLM) oversized scrubbing filters would be required to provide sufficient residence times.

**"L 125 / L 144:** What is the PMT temperature? Is the PMT actively cooled? What causes the background signal? Would the background decrease, if the PMT was cooled down at temperatures below 0°C by an active cooling unit?"

**Response:**

The internal temperature of the PMT is 0°C. It is regulated by a built-in thermoelectric cooler and this feature is part of the hardware from the manufacturer. We only have the control to regulate the surface temperature of the PMT. This is done externally by using a water chiller at 20°C or 25°C (avoiding condensation) according to manufacturer recommendations. The dark counts on the PMT signal are generally less than 50 counts $s^{-1}$ for the channels used for the $NO_2$ fluorescence detection. The major reason for the background signal, larger than the dark signal typically by a factor >25, is expected to be fluorescence contamination from the Herriot cell mirrors existing in the red region of wavelength.

**"L 223:** ". . . at a lower temperature . . ." Which temperature?"

**Response:**

The sentence was to give a general statement.

Under lower temperature conditions, the reaction between NO and $O_3$ slows down. This can lead to a change in the conversion efficiency from NO to $NO_2$. In our case, many electrical parts (electronic valves, ozone generator, and mass flow controllers) are installed inside the calibration unit. In a fully operational mode for one day, the temperature build up in the calibration unit is 8-10°C higher than ambient temperatures. From our experience/observations, conditions with a temperature lower than 20°C inside the calibrator do not occur.

**"L 237:** Is there an explanation for the change in sensitivity? What is the range of sensitivity change?"

**Response:**

Generally, some factors can contribute to a change in the sensitivity e.g., stability of the optics alignment, cleanness of the optics, temperature related effect of electronics, stability of the calibration signal etc. Frequent calibrations were performed during the PARADE-2011 to assess the stability of the sensitivity. Based on calibrations (> 130) performed during PARADE-2011 by using dry-air (< 25 ppm of water), the relative variation in the sensitivity of the instrument was better than ± 2.7 % (1σ). Further, the sensitivity of the instrument decreases by 5 % (relative to the dry-air) at 1 % of atmospheric $H_2O$ vapour. This is corrected by using simultaneous measurement of $H_2O$ vapour.

**"L 268:** Figure 8 shows the relative precision obtained during different calibrations. But how exactly do you determine the relative precision? Does it include for example only the variability of the sensitivity or the variability of the background, etc.? Please describe in more detail."

**Response:**

The text in this section of the draft is simplified and the figure is now presented as a function of $NO_2$ mixing ratios. The relative precision 0.5 % (1 min$^{-1}$) was calculated based on the standard deviation of the PMT $NO_2$-signal (in s$^{-1}$ time resolution) during the calibration period for different $NO_2$ concentrations. Since the selected signals were based on higher levels of $NO_2$ concentrations (> 0.5 ppb). So the number 0.5 % (1 min$^{-1}$) is true representative of precision at higher $NO_2$ concentrations.

Standard deviation of signals at different $NO_2$ concentrations can be extrapolated to zero for determination of the precision at background level. It can also be calculated from the standard deviation of the zero-air signal. Both approaches give a similar result of about 3 ppt precision for our instrument. Hence, the total precision was defined by considering the both values i.e., 0.5% (1 min$^{-1}$) + 3 ppt (1$\sigma$).

**"L 318:** R6 is not a valid chemical reaction (both sides of the reaction arrow should be balanced)."

**Response:**

It is modified as follows

$1^{st}$ step: $NO_3 + h\upsilon_{DiodeLaser} \rightarrow O + NO_2$ $\qquad$ $2^{nd}$ step: $NO_2 + h\upsilon_{DiodeLaser} \rightarrow NO_2^* \rightarrow NO_2 + h\upsilon$ $\qquad\qquad$ R. 1

**"L 337:** I agree the short residence time of the sample air inside the instruments minimizes the thermal decomposition of the respective species. But please give at least a few calculated lifetimes against thermal decomposition for the most important interfering gases that illustrate this statement."

**Response:**

At this line, the discussion was referring to the sampling line prior to the orifice. This information is further clarified in the text.

The ambient lifetime based on thermal decomposition is added for different species. The lifetime inside the instrument would be much larger as the cell pressure is about a factor 100 smaller compared to the ambient pressure.

**"L505:** I suggest that in the summary the authors underline the main advantage and disadvantage of this measuring system, also in comparison with other measurement techniques. What is the future of Gandalf (besides LOTR)? Are there specific plans to use this instrument during other field campaigns?"

**Response:**

Following "sentences" are included in the summary of the draft.

"In general, all instruments performed well. GANDALF showed a very good correlation (R$^2$ ≈ 0.99) in comparison to other in situ instruments (Fig. S11 in the supplement) and even with LP-DOAS the correlation was R$^2$ ≈ 0.9. The differences in the absolute values were within the specified range of individual measurement errors. The main advantages and disadvantages of GANDALF compared to the other instruments are summarized as follows.

[revised manuscript text omitted]

"**L 508:** The authors are mentioning that the instrument is capable for measurements throughout the troposphere with a time resolution of 1 s to 1 min. However, the whole preceding discussion has been focused on ground based measurements at a time resolution of 1 min. Also the concentration of NO2 in the free troposphere is much lower than in the boundary layer. LOD would increase significantly if you reduced the sampling time from 60 s to 1s. Please
outline in short what improvements would be necessary to achieve this goal."

**Response:**

$NO_2$ in the free troposphere is variable (seasonally) and generally lower than 50 ppt (Gil-Ojeda et al., 2015). Depending on the location, in the free troposphere and the marine boundary layer, $NO_2$ can be as low as a few ppt (Beygi et al., 2011;Schreier et al., 2016). These $NO_2$ ranges are below the detection limit for the instrument for short time resolutions of 1s, for example.
Improvements for future use on aircraft are possible by further reducing the background of the instrument. Since most of the background signal is from the fluorescence contamination of the Herriot's cell mirrors, this could be avoided by using a single beam (as demonstrated by (Di Carlo et al., 2013)) of the laser for detection without a Herriott cell or by using different coatings on the Herriott cell mirrors to increase reflectivity and reduce fluorescence.

**"Technical Corrections:"**

**"L 298:** . . . about 8 time higher than the cross section of . . ."
**Response:**
It is done.

**"Tables:"**

**"Table 3:** ±δ – explanation in the caption is missing."
**Response:**
It is done.

**"Table 4:** Uncomplete caption - which ratios?"
**Response:**
It is modified as follows.
"Average the ratios of $NO_2$ measurements from the different instruments, taking into account all available data from PARADE-2011."

**"Figures:"**
"In general the figure captions are often not sufficient in explaining the content of the figures."
**"Figure 1:** The numbers in the caption of this figure have different orientations and do not facilitate the reading. All numbers should have the same conventional orientation (like the numbers "9, 10, . . . ."?. "SF" - This is not quite
consistently. All other objects of this figure stand for units of the instrument. "Sampling flow" is the gas stream into the instrument (I assume) and not part of the instrument. So it would be more suitable to write: Inlet orifice or sampling flow line, or . . . ."
**Response:**
The numbering in the figure/text is simplified, and synchronized.

**"Figure 4:** . . . as a function of O3 concentration in . . . Please explain "arb" in the y-axis label. The caption is incomplete; "Box-Model NO2" is not mentioned."
**Response:**
It is modified as follows.

"The PMT NO$_2$ signals in counts (cts) are shown as a function of O$_3$ concentrations in the calibrator (y-axis scale on the left side), together with NO$_2$ calculated from a box model of the NO$_2$ production in the calibrator (y-axis scale on the right side)."

**"Figure 5:** "also theoretically calculated residence time (7.73s). . ." I assume the red line in this figure is meant."

**Response:**

Yes, the red line is showing the theoretical residence time. This information is now added to the caption in the draft.

**"Figure 6:** What do you mean by calibration signal? I assume it is the number of counts at the PMT?"

**Response:**

Yes, these are counts at the PMT. It is corrected in the caption.

**"L 250 – L 265 / Figure 7:** The description in the text and in the figure caption is a little bit confusing and should be clarified. E.g. an ozone analyzer is shown in the figure but not mentioned in the text. In the text blue, red and white arrows are mentioned; in the figure you find additionally orange arrows. In the text only red arrows in L2 are mentioned, but there also white arrows found in L2. I assume that the valves EV3 and EV2 have to point at the position P1(P2) at the same time? Above the Gandalf-box in Figure 7 there are three times written "4100 sscm" in different colors and different orientation. As long as you do not discuss it explicitly in the text, one "4100 sccm" label is enough. Figure caption: "outdoor – operations"? - Better during field campaigns or during the PARADE field campaign."

**Response:**

The ozone analyser is used to check the concentration of O$_3$ in the calibration gas, and this information is now included in the text. The orange arrow has been removed. The white arrow was representing ambient air flow during along the overflow of the calibration gas. P1 and P2 where switched around. The figure has been revised along with the text to correct it and make it easier to understand in the updated version of the draft.

**"Figure 8:** JD = Julian Days. The formulation of this caption is a little bit unclear. Please improve."

**Response:**

The relative precision in this figure is now shown as a function of NO$_2$ mixing ratios instead of time. The caption of the figure is also accordingly changed.

**"Figure 12 and 13:** "Ratios. . ." – The readability would be improved if you would write in the caption which ratio is meant. Please choose the same y-scale for all figures."

**Response:**

All the figures are modified for as suggested by the reviewer.